# INSTRUCTION MINING: INSTRUCTION DATA SELECTION FOR TUNING LARGE LANGUAGE MODELS

## ABSTRACT

Large language models (LLMs) are initially pretrained for broad capabilities and then finetuned with instruction-following datasets to improve their performance in interacting with humans. Despite advances in finetuning, a standardized guideline for selecting high-quality datasets to optimize this process remains elusive. In this paper, we first propose INSTRUCTMINING, an innovative method designed for automatically selecting premium instruction-following data for finetuning LLMs. Specifically, INSTRUCTMINING utilizes natural language indicators as a measure of data quality, applying them to evaluate unseen datasets. During experimentation, we discover that double descent phenomenon exists in large language model finetuning. Based on this observation, we further leverage BLENDSEARCH to help find the best subset among the entire dataset (i.e., 2,532 out of 100,000). Experiment results show that INSTRUCTMINING-7B achieves state-of-the-art performance on two of the most popular benchmarks: LLM-AS-A-JUDGE and Huggingface OPENLLM.

## 1 INTRODUCTION

Large language models (LLMs) have demonstrated transformative capabilities, powering numerous applications with the strong ability in automatically generating responses according to human instructions (Ouyang et al., 2022; Peng et al., 2023; Chung et al., 2022; Scao et al., 2022). However, it is hard sometimes for language models to capture the meaning of human instructions and respond to them even if they are pretrained with large amount of data. To counter this challenge, instruction tuning emerged as a paramount method in tailoring the behaviours of LLMs, which leverages instruction-following pairwise data (i.e., instruction data) during finetuning (Wei et al., 2021; Ouyang et al., 2022; Chung et al., 2022; Wang et al., 2022a). A recent study LIMA has revealed that even a small amount of carefully selected high-quality instruction data can significantly improve model performance through instruction tuning (Zhou et al., 2023). Nevertheless, LIMA still requires human experts to filter examples from extensive datasets, which is both time-consuming and expensive.

In this paper, we propose INSTRUCTMINING, a novel method designed to automatically select high-quality instruction data for finetuning better LLMs. Achieving this objective necessitates a data evaluator capable of assessing the quality of instruction data without the intervention of human experts. Furthermore, the data selector is also indispensable for automatically identifying the most suitable subset of instruction data for finetuning LLMs. Nevertheless, quantifying the quality of instruction data without human expert is a non-trivial task. To address this problem, we employ the loss incurred by a finetuned model on the evaluation set a proxy for data quality. However, computing this inference loss necessitates the actual finetuning of a language model, a potentially time-consuming process. To overcome this obstacle, we introduce a set of selected natural language indicators (e.g., reward model score), capable of predicting the inference loss without the need for finetuning an LLM. This approach can efficiently provide an estimation of the dataset's quality.

While our approach can assess and rank the entire dataset based on data quality, determining the most optimal subset for finetuning LLMs remains an unresolved challenge. A straightforward solution is to choose the top-$K$ high-quality data samples, but selecting the appropriate $K$ proves to be inherently difficult in practice. To address this complexity, we conducted a series of experiments exploring the relationship between data quantity and quality. Remarkably, as we continued to increase the subset size for finetuning language models, we observed the double descent phenomenon (Nakkiran et al., 2021), as illustrated in Figure 1. This observation signifies a transition in the primary determinant of

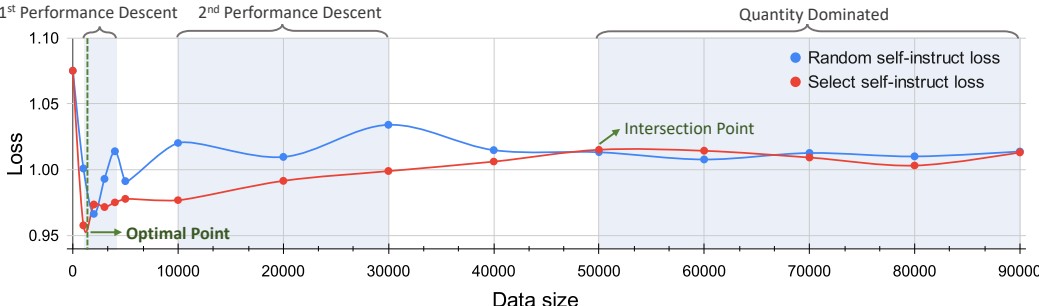

Figure 1: Double descent phenomenon in generative language models. Higher loss indicates worse performance. Red line refers to INSTRUCTMINING selected data sizes w.r.t. model inference loss. Blue line refers to random selected data sizes w.r.t. model inference loss. Our optimization goal is to find the optimal point which results in the lowest inference loss.

model performance from data quality to data quantity once the data size crosses a specific threshold. In such scenarios, focusing on an initial set of high-quality data points (e.g., $K$=10,000) is more efficient for identifying the optimal point than perusing the entire dataset. Given the cost-sensitive nature of pinpointing this optimal point, we employ BLENDSEARCH (Wang et al., 2021a) to automatically search for the best subset for our needs.

We further substantiate the validity and scalability of INSTRUCTMINING by contrasting its performance with other state-of-the-arts across diverse benchmarks. Notably, INSTRUCTMINING enhances the performance of LLAMA-2-7B by 4.93 on the Huggingface OPENLLM benchmark. In addition, our finetuned models are able to generate equivalent or superior responses in 64.67% of instances, compared to VICUNA-7B-v1.5. Furthermore, INSTRUCTMINING contributes to heightened finetuning efficiency. The optimal INSTRUCTMINING model, finetuned on a mere 2.5% (i.e., 2,532 out of 100,000) of the highest-quality examples from the complete dataset, can achieve state-of-the-art on both LLM-AS-A-JUDGE (Zheng et al., 2023) and OPENLLM benchmarks.

Our contributions are summarized as follows:

- In this work, we pioneer the application of classical data mining techniques to enhance LLMs by autonomously selecting high-quality data. To realize this objective, we introduce INSTRUCTMINING, a method encompassing data assessment and selection processes.
- The proposed INSTRUCTMINING innovatively combines customized language indicators with an advanced searching algorithm, enabling the automatic assessment of data quality and identification of the optimal subset for finetuning language models.
- Models finetuned with INSTRUCTMINING exhibit state-of-the-art performance on two of the most popular benchmarks: LLM-AS-A-JUDGE and Huggingface OPENLLM. Meanwhile, Utilizing less training data can effectively reduce both the training time and cost.

## 2 METHODOLOGY

In this section, we provide a detailed description of our proposed method, INSTRUCTMINING. A procedure graph is provided in Figure 5. Our method is composed of two parts, quality estimation and threshold search. We first introduce our method for estimating instruction data quality in Section 2.1. This is achieved by aligning the data quality with the inference loss of a fine-tuned model. Then we propose our evaluation rule along with the selected natural language indicators in Section 2.2. Finally, we present the observed double descent phenomenon and introduce a BLENDSEARCH-based data selector in Section 2.3.

### 2.1 WHAT IS INSTRUCTION QUALITY?

In this paper, we follow the superficial alignment hypothesis proposed by Zhou et al. (2023) that a model's knowledge is mostly learnt during pretraining, while instruction data teaches the model to follow a certain pattern when interacting with users. Hence, the quality of these instruction data

| Indicator | Notation | Explanation |
|---|---|---|
| Input length | $Len_{in}$ | The number of tokens in tokenized inputs. |
| Output length | $Len_{out}$ | The number of tokens in tokenized outputs. |
| Reward score | $Rew$ | The `oasst-rm-pythia-1.4b` reward model inference score of every pair in the dataset. (Köpf et al., 2023) |
| Perplexity | $PPL$ | The exponentiated average negative log-likelihood of response. |
| MTLD | $MTLD$ | Measure of Textual Lexical Diversity (McCarthy & Jarvis, 2010) |
| KNN-i | $KNN_i$ | Distance to approximate $i^{th}$-nearest neighbors (Dong et al., 2011) in SentenceBERT(Reimers & Gurevych, 2019) embedding space. |
| Unieval-naturalness | $Nat$ | The score of whether a response is like something a person would naturally say, provided by the UniEval (Zhong et al., 2022) dialogue model. |
| Unieval-coherence | $Coh$ | The score of whether this response serves as a valid continuation of the previous conversation, provided by the UniEval (Zhong et al., 2022) dialogue model. |
| Unieval-understandability | $Und$ | The score of whether the response is understandable, provided by the UniEval (Zhong et al., 2022) dialogue model. |

Table 1: Summary of indicators for instruction quality evaluation. Each data sample is viewed as a pair of instruction and response (i.e., input and output) of LLM.

could be viewed as its ability to efficiently steer language models in learning to generate responses in a particular manner. Based on this assumption, we further propose our instruction quality evaluation hypothesis as follows.

**Hypothesis 1** *Instruction Quality Evaluation Hypothesis: Given an instruction dataset $D$, we finetune a language model on $D$, denoted as $M_{ft}$. The instruction quality of $D$ can be estimated through the inference loss of $M_{ft}$ on a evaluation dataset $D_{eval}$.*

To ensure the inference loss provides a valid measure for evaluating data quality, the evaluation set should comprise a selected collection of unbiased and high-quality instruction-following samples.

In particular, given an instruction-following dataset $D$, we finetune a base language model $M$ using $D$ with model training settings $S$. $S$ normally refers to training batch size, epochs, etc. $L$ refers to the loss function. The obtained finetuned language model is denoted as $M_{ft}$. We define the dataset $D$'s quality $Q_{D|M,S}$ as below,

$$Q_{D|M,S} \propto -L(M_{ft}, D_{eval}) \tag{1}$$

where $D_{eval}$ refers to the high-quality and unbiased evaluation set, and $\propto$ means a direct proportion.

## 2.2 HOW TO ESTIMATE INSTRUCTION QUALITY?

According to Equation 1, we utilize the inference loss to evaluate instruction quality. However, finetuning an LLM for evaluation can be inefficient, since this process can take days of training. To solve this problem, we introduce a set of natural language indicators and use the indicators to predict the inference loss. In this paper, We have a set of indicators $I = \{I_i, i = 0, \cdots, n\}$, summarized in Table 1. For a given instruction dataset $D$, we compute the corresponding indicator values $I(D) = \{I_i(D), i = 0, \cdots, n\}$. There exists a function $F$ such that the aforementioned model inference loss $L(M_{ft}, D_{eval})$ can be approximated using $F(I(D))$.

The relationship between the finetuned model inference loss $L$ and these computed indicators can be formulated as in Equation 2.

$$\underbrace{-Q_{D|M,S}}_{\text{Instruction Quality}} \propto \log \underbrace{L(M_{ft}, D_{eval})}_{\text{Model Evaluation Loss}} \simeq \underbrace{L_0}_{\text{Minimal Loss Constant}} + F\{I_1(D), I_2(D), \cdots, \underbrace{I_i(D)}_{i\text{th Indicator on data } D}, \cdots, \underbrace{I_n(D)\}}_{\text{Bag of Indicators}} \tag{2}$$

In this paper, we assume that there exists a multivariate linear function of $I$ that is proportional to the logarithmic loss. Consequently, Equation 2 can be reparameterized as Equation 3:

$$\begin{aligned} \log L(M_{ft}, D_{eval}) &\propto L_0 + F\{I(D)\} \\ &\propto L_0 + \beta_0 + \beta_1 I_1(D) + \beta_2 I_2(D) + \cdots + \beta_n I_n(D) + \epsilon \end{aligned} \tag{3}$$

where $\beta_0$ denotes the linear constant, $\beta_i, i \in \{1, \cdots, n\}$ represents a sequence of linear coefficients, and $\epsilon$ refers to the random error term.

To investigate the relationship between these indicators and the overall dataset quality, it becomes necessary to accumulate experimental results to estimate the unknown parameters $\beta_i, i \in \{0, \cdots, n\}$. In this study, we employ the Least Squares method (Björck, 1990) to estimate the parameters in the multivariate function. The Least Squares method is a standard approach in regression analysis for the approximate solution of overdetermined systems. The technique minimizes the sum of the square residuals, thus providing the optimal fit between the observed and predicted data in terms of reducing the overall prediction error. Our experimental results and analysis are detailed in Section 4.

### 2.3 INSTRUCTION DATA SELECTOR

During experimentation, we observe that with larger data size, model performance first gets better and then gets worse. After the data size grows to a certain level, model performance gets better again. Further analysis are provided in Section 5.1. This phenomenon indicates that there is an optimal point where better performance can be obtained with a smaller amount of data. Hence, searching for the best data size is important for finetuning a better language model.

To achieve our objective, we employ BLENDSEARCH (Wang et al., 2021a) in Flaml (Wang et al., 2021b) library to determine the optimal data size. BLENDSEARCH effectively combines global and local optimizations by Bayesian optimization and different local search threads, making it efficient for searching cost-related hyperparameters and complex search spaces with local optima. In our context, we leverage a logarithmic uniform distribution to randomly sample the dataset size, treating the dataset size as the experiment's cost since the training time scales proportionally with the dataset size. The search goal is to minimize the loss on the evaluation set.

## 3 EXPERIMENT SETTINGS

Our experiments mainly focus on two goals. The first goal is to estimate the unknown parameters in the proposed INSTRUCTMINING rule. The second one is to evaluate and analyze the performance of INSTRUCTMINING over varied finetuning scenarios. Section 3.1 elaborates rule estimation empirical study design. Section 3.2 details the datasets we use for conducting both estimation and evaluation experiment. Section 3.3 elaborates the finetuning settings we used for estimation and evaluation.

### 3.1 EMPIRICAL EXPERIMENT DESIGN

The general procedure of our rule estimation experiment is shown in Figure 5. To estimate the correlation between the evaluation loss and indicators $I$, we need to get datasets of different indicator values. To achieve this, we first select several commonly used datasets with different presumed quality levels and fuse them together with randomly sampled percentages to create finetune datasets. These sampled finetune datasets should encompass varying proportions of preassumed high quality and low quality examples. For each of these sampled datasets $D_i$, we compute its respective indicator values $I(D_i)$ and finetune a base language model $M$ using $D_i$. Following Equation 1, the quality $Q_{D_i}$ for dataset $D_i$ is approximated using the evaluation loss of finetuned model $M_{ft,i}$ on a fair evaluation dataset $D_{eval}$. Following the collection of a range of results correlating $Q_{D_i}$ with $I(D_i)$, we undertake a statistical regression analysis to discern relationships within the dataset.

### 3.2 DATASETS

**Candidate datasets for rule fitting.** In order to create diverse training datasets, we collect data from various sources. This approach ensures that the datasets exhibit differences in quality and maintain diversity among sources. For this purpose, we have selected the following datasets as candidate

| Datasets | Sourced from | Size | Quality | Usage |
|---|---|---|---|---|
| ALPACA | Generated w/ davinci | 52.0k | Normal | Est. Candidate |
| OPEN-ASSISTANT | human-generated | 3.4k | Both | Est. Candidate |
| STACKEXCHANGE | human-generated | 3.0k | High | Est. Candidate |
| WIKIHOW | human-generated | 2.0k | High | Est. Candidate |
| DOLLY | human-generated | 15.0k | Normal | Evaluation |
| OPENORCA | Generated w/ GPT-4 | 1M | High | Evaluation |
| OPENORCA | Generated w/ GPT-3.5 | 3M | Normal | Evaluation |

Table 2: Overview of datasets used during experiment.

datasets: ALPACA (Taori et al., 2023), OPEN-ASSISTANT (Köpf et al., 2023), STACKEXCHANGE, and WIKIHOW. Due to the varying formats, sizes, and distributions of different datasets, we have applied distinct processing procedures to each dataset. Table 2 provides an overview of the candidate training datasets after preprocessing. As mentioned in section 3.1, we merged candidate training datasets, resulting in each dataset containing 1,000 instruction-output pairs. We generated a random number $r_i$ for each dataset and randomly selecting $1000 * r_i / \sum_i r_i$ samples from each dataset for combination. Besides, considering the significant size difference between ALPACA and other candidate datasets, we randomly sampled 2,000 data examples from ALPACA to maintain scale consistency across all the candidate datasets.

**Test set for rule fitting.** To address real-world requirements, we use the SELF-INSTRUCT dataset Wang et al. (2022a), which contains 252 instructions as rule-fitting test set. Considering evaluation efficiency, we randomly sampled 80 instructions from the whole dataset as our evaluation set. In our study, we employed `gpt-4` from OPENAI to generate response for each instruction.

**Datasets for rule evaluation.** We further test INSTRUCTMINING by using it to select high-quality examples from unseen datasets for finetuning large language models. During evaluation, we mainly use OPENORCA and DOLLY-15K as the two unseen candidate datasets. For OPENORCA, given its extensive size, we randomly select 50,000 examples from OPENORCA-GPT3.5 and 50,000 examples from OPENORCA-GPT4 for experimentation (henceforce refered to as OPENORCA). To make sure that our method does not overfit on the SELF-INSTRUCT evaluation set, we use the `gpt-4` labeled MT-BENCH dataset (Zheng et al., 2023) as an unseen evaluation set. To be noticed, since our candidate and evalation datasets does not include multi-turn examples, when evaluating on MT-BENCH, we only use the first turn of MT-BENCH dataset.

### 3.3 FINETUNING SETTINGS

We conduct all finetuning experiments on the same base model LLAMA-2-7B (Touvron et al., 2023). All finetuning datasets during estimation phase are of the same size, 1,000 examples in each. We run model finetuning for 3 epochs, with per step batch size set to 128. We use Adam with $\beta_1 = 0.9, \beta_2 = 0.999$, and cosine learning rate scheduler starts from $2e - 5$, decays to 0. Each model finetuned during estimation phase is evaluated on the evaluation dataset mentioned in section 3.2. We run all finetuning and evaluation experiments on a NVIDIA A100 80G GPU cluster, with 8 A100 GPUs used in each experiment.

## 4 EXPERIMENTAL RESULTS

### 4.1 INSTRUCTMINING PARAMETER FIT

Following Section 3.1, we randomly sampled 129 subsets from the entire data pool with different percentages. These subsets are then used to finetuned 129 corresponding language models. We collect the inference loss and indicator values for each subset. To select the optimal rule, we first choose regression results with the highest $R^2$, and then prioritize the rule with the most significant $p$ values. The detailed regression result is available in Table 9. Based on this result, we delineate our estimated evaluation function, which is articulated as Equation 4. Accordingly, reward score and unieval scores appear to be the most significant indicators in the quality rule. This estimation result reveals that $Und$ is in negative correlation with data quality, while the other three indicators are of

| Dataset | Sampling Method | Total Time(min) | Rule | Data Size | Loss | Loss(MT-BENCH) |
|---------|-----------------|-----------------|------|-----------|------|----------------|
| OpenOrca | *Selected* | 150(Rule)+15(Train) | -0.1347 | 1,000 | **0.958** | 0.711 |
| | | 150(Rule)+300(Train) | -0.0716 | 20,000 | 0.991 | 0.730 |
| | | 150(Rule)+1350(Train) | -0.0243 | 90,000 | 1.014 | 0.735 |
| | ***BlendSearch*** | **150(Rule)+35(Train)** | -0.1197 | **2,532** | 0.973 | **0.699** |
| | *Random* | 15(Train) | -0.0195 | 1,000 | 1.001 | 0.746 |
| | | 300(Train) | -0.0180 | 20,000 | 0.991 | 0.751 |
| | | 1350(Train) | -0.0176 | 90,000 | 1.010 | 0.763 |
| Dolly | *Selected* | 22(Rule)+15(Train) | -0.0969 | 1,000 | 1.0429 | 0.7964 |
| | | 22(Rule)+75(Train) | -0.0622 | 5,000 | 1.0327 | 0.7847 |
| | | 22(Rule)+150(Train) | -0.0449 | 10,000 | 1.0371 | 0.8001 |
| | ***BlendSearch*** | **22(Rule)+35(Train)** | -0.0770 | **2,648** | **1.0160** | **0.7746** |
| | *Random* | 15(Train) | -0.0286 | 1,000 | 1.0409 | 0.8215 |
| | | 75(Train) | -0.0289 | 5,000 | 1.0331 | 0.7910 |
| | | 150(Train) | -0.0293 | 10,000 | 1.0356 | 0.8086 |

Table 3: Quality-guided instruction selection experiment result. Rule refers to the average of our quality rule score on the dataset. *Selected* $k$ data refers to the top $k$ data examples with the highest quality scores. We calculate the inference loss values over two evaluation sets, SELF-INSTRUCT and MT-BENCH. Total time refers to the time spent during feature extraction, rule calculation, and finetuning using 8 GPUs.

positive correlation with data quality.

$$Q_{D|M,S} \propto -L(M_{ft}, D_{eval})$$
$$\log L(M_{ft}, D_{eval}) \propto 0.0274 - 0.0078Rew + 0.4421 * Und - 0.3212 * Nat - 0.1520 * Coh + \epsilon \tag{4}$$

### 4.2 QUALITY-GUIDED INSTRUCTION SELECTION

We follow the estimated INSTRUCTMINING rule in Equation 4 to select high quality examples from two unseen datasets, OPENORCA (Lian et al., 2023) and `databricks-dolly-15k` [1]. The experiments are all based on LLAMA-2-7B model. We first elaborate our BLENDSEARCH results. Then we present our evaluation results of various model finetuned during this process. These models are first evaluated on two evaluation sets, SELF-INSTRUCT and MT-BENCH. The searched best finetuned model is then assessed using LLM-JUDGE and OPENLLM benchmarks.

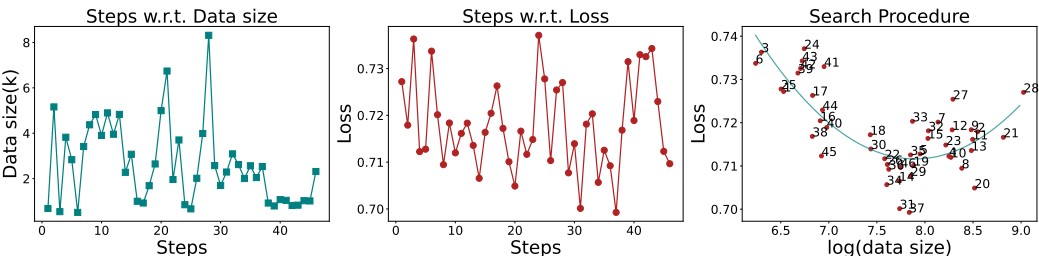

Figure 2: BLENDSEARCH results. The Loss is calculated on MT-BENCH evaluation set.

**BlendSearch results.** In response to the performance decline-rise phenomenon with increasing training data, we conduct a BLENDSEARCH within a range of data sizes from 512 to 10,000. Our prior empirical findings suggest that a maximum dataset size of 10,000 is sufficient for optimizing the data size. Figure 2 details the search procedure from the perspective of steps.

**Loss results.** We first evaluate the finetuned models using inference loss on SELF-INSTRUCT and MT-BENCH dataset. Results are presented in Table 3. According to the results, INSTRUCTMINING

---

[1]https://www.databricks.com/blog/2023/04/12/dolly-first-open-commercially-viable-instruction-tuned-llm

| Model | Data size | Avg. Metric | ARC | HellaSwag | MMLU | TruthfulQA |
|---|---|---|---|---|---|---|
| INSTRUCTMINING-Selected | 10,000 | 58.65 | **56.66** | 79.77 | 49.89 | 48.26 |
| INSTRUCTMINING-Selected | 40,000 | 59.25 | 54.44 | **80.11** | 52.60 | 49.83 |
| INSTRUCTMINING-Random | 10,000 | 58.74 | 54.78 | 79.58 | 49.02 | **51.58** |
| INSTRUCTMINING-Random | 40,000 | 58.95 | 54.78 | 79.89 | 51.16 | 49.95 |
| VICUNA-1.5-7B | 125,000 | 57.99 | 53.24 | 77.39 | 51.03 | 50.33 |
| LLAMA-2-7B-chat | 27,540+ | 56.34 | 52.90 | 78.55 | 48.32 | 45.57 |
| LLAMA-2-7B | - | 54.32 | 53.07 | 78.59 | 46.87 | 38.76 |
| STABLEBELUGA-7B | 600,000 | **59.59** | 56.31 | 79.14 | **52.71** | 50.19 |

Table 4: OPENLLM benchmark scores. We use the same evaluation setting as OPENLLM leaderboard. For ARC benchmark, we use 25 few shots. For HELLASWAG, we use 10 shots. For MMLU, we use 5 shots. For TRUTHFULQA, we use zero shot.

can efficiently select high-quality data from various unseen datasets. Full data finetuning on LLAMA-2-7B with OPENORCA dataset can take up to 30 hours of 8 GPU time. With INSTRUCTMINING, we are able to select the top 1,000 data examples in around two hours and training a better LLM within 15 minutes. This is also valid with DOLLY. In addition, we discover that despite different sampling methods, the loss values on both SELF-INSTRUCT and MT-BENCH always tend to increase with larger data sizes. To study this phenomenon, we designed further experiments to investigate the relationship between finetuning data size and LLM performance. Details are provided in section 5.1.

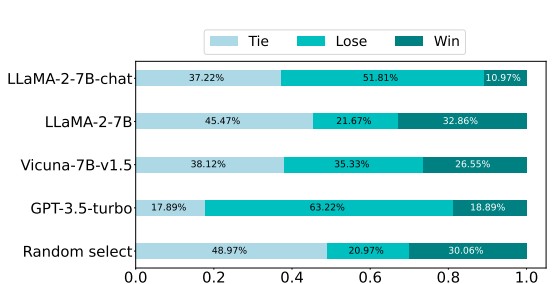

(a) GPT-4 preference evaluated results. Tie means GPT-4 assesses two responses as equal. Lose means GPT-4 prefers the other model's response. Win means GPT-4 prefers INSTRUCTMINING model response.

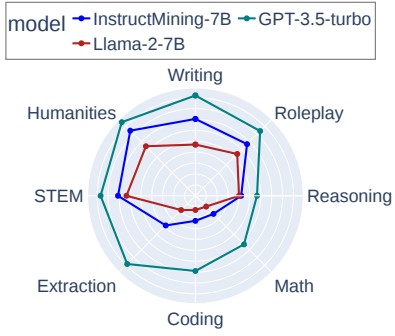

(b) GPT-4 assessed model ability result. We prepared tasks from different categories in MT-BENCH, and let GPT-4 to evaluate the generated response.

Figure 3: LLM assessed results.

**LLM assessment.** We next compare our best model with other state-of-the-art models using LLM-AS-A-JUDGE. We let GPT-4 choose the better answer between two responses generated by two different models. According to the results presented in Figure 3, our model is able to generate better or equal results in 64.67% of the cases compared to VICUNA-1.5-7B. We also let GPT-4 assess the model from different perspectives. According to Figure 3b, our model significantly improves the original LLAMA-2 model's ability in writing, roleplay, humanity, STEM, extraction and coding.

**OpenLLM benchmark results.** Besides, we further test our finetuned models on the widely used OPENLLM benchmark (Gao et al., 2021). OPENLLM benchmark is composed of four widely used general question answering benchmarks, ARC (Clark et al., 2018), HELLASWAG (Zellers et al., 2019), MMLU (Hendrycks et al., 2020) and TRUTHFULQA (Lin et al., 2021). During experimentation, we align our inference settings with huggingface OPENLLM leaderboard settings. Results are available in Table 4. Notably, INSTRUCTMINING finetuned models can achieve similar performance compared to STABLEBELUGA-7B, which is the state-of-art LLAMA-2-7B based model on OPENLLM leaderboard. Furthermore, INSTRUCTMINING only requires around two hours of indicator inference and ten hours of finetuning to get a comparably strong language model. We also discover that, when evaluating with some metrics, larger data does not always promise better

| Rew | Und | Nat | Coh | Loss(SELF-INSTRUCT) | Loss(MT-BENCH) |
|-----|-----|-----|-----|---------------------|-----------------|
| ✓ | ✓ | ✓ | ✓ | 0.958 | 0.711 |
| ✗ | ✓ | ✓ | ✓ | 0.988 (↑0.030) | 0.762 (↑0.051) |
| ✓ | ✗ | ✓ | ✓ | 0.989 (↑0.031) | 0.746 (↑0.035) |
| ✓ | ✓ | ✗ | ✓ | 0.977 (↑0.019) | 0.742 (↑0.031) |
| ✓ | ✓ | ✓ | ✗ | 0.969 (↑0.011) | 0.742 (↑0.031) |
| ✗ | ✗ | ✗ | ✗ | 1.001(↑0.043) | 0.746(↑0.035) |

Table 5: Ablation study result. All results are compared with the original INSTRUCTMINING rule. The final row refers to unfiltered randomly selected data.

performance. For instance, accuracy on ARC tends to decrease when the data size increases. Further analysis of this phenomenon is provided in section 5.1.

## 4.3 ABLATION STUDY

We further conduct ablation experiments to study the influence of every indicator in INSTRUCTMIN-ING. To do this, we first remove one indicator from the current rule and estimate a new rule using the other three indicators, based on the original random experiment results. Then, we use this new rule to select 1,000 data examples with the highest scores. These 1,000 data examples are later used to finetune the base language model, LLAMA-2-7B, for three epochs. We present ablation study result in Table 5. Accordingly, $Rew$ appears to be the most important indicator among the four. Without $Rew$ as one of the rule indicators, the estimated rule results in an increase of 0.03 in SELF-INSTRUCT inference loss and 0.051 in MT-BENCH inference loss.

## 5 ANALYSIS

### 5.1 DOUBLE DESCENT IN GENERATIVE LANGUAGE MODELS

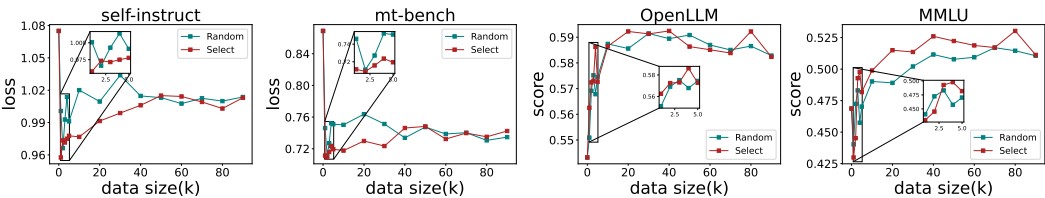

Figure 4: Double descent in generative language models. Models are evaluated using four metrics: loss on SELF-INSTRUCT, loss on MT-BENCH, OPENLLM scores and MMLU scores..

In this section, we present further experimental findings on `OpenOrca` dataset. In previous experiments, we find out that a language model's performance can be influenced by both finetuning data quality and quantity. When data quantity increases, generative language models' performance does not promise to become better. This phenomenon suggests a balance between data quantity and data quality. Results are presented in Figure 4. This reveals some interesting emergent phenomena when finetuning large language models. We detail the observed phenomena below.

**Phenomenon 1** *Non-monotonicity exists in language model performance. As we increase the training data size, language model performance first becomes better then gets worse. When data size increases to a certain level, performance becomes better again.*
Based on Figure 4, we observe that the performance first improves as the training data size grows. Then, after the data size grows to around 10,000, loss begins to increase, meaning that language model performance worsens. Finally, as data size continues to grow, language model's performance improves. This phenomenon is similar to the double descent phenomenon (Nakkiran et al., 2021) that non-monotonicity exists with varying numbers of training samples. In our experiment, we observe that this phenomenon not only exists in vanilla language model training but also in large generative language model finetuning.

**Phenomenon 2** *Balance point exists between randomly selected and quality selected data. As data size grows, data quality becomes a less important factor for model performance.*

Given Figure 4, we find out that when data size grows to a certain point, the performance curve of selected data will always intersect with the random one. Besides, the distance between the two decreases as the data size increases. This phenomenon indicates that data quality measure can help improve model performance at first. However, data quality becomes less important when data size grows to a certain level.

## 5.2 ROBUSTNESS

To further explore the effectiveness of INSTRUCTMINING, we evaluate it across three different finetuning settings: *(1) Different base models.* We change the original base model LLAMA-2-7B into LLAMA-1-7B to test whether our method is scalable to other models. *(2) Different model sizes.* We change the original 7B model size into 13B to test whether our method is scalable to other model sizes. *(3) Parameter efficient settings.* LORA (Hu et al., 2021), a parameter efficient method, is widely used when finetuning a large language model to help save GPU memory usage. We also test our method with LORA settings to see whether INSTRUCTMINING is scalable to parameter efficient finetuning. Results are presented in Table 6. As the data shows, INSTRUCTMINING rule can be applied to various base models, model sizes and parameter efficient settings.

| Base Model | Model Size | LoRA | Sampling Method | Loss(SELF-INSTRUCT) | LOSS(MT-BENCH) |
|---|---|---|---|---|---|
| LLAMA-2 | 13B | ✗ | *Selected* | **0.8748** | **0.6531** |
| | | | *Random* | 0.8983 | 0.6589 |
| LLAMA-1 | 7B | ✗ | *Selected* | **1.013** | **0.798** |
| | | | *Random* | 1.056 | 0.844 |
| LLAMA-2 | 7B | ✓ | *Selected* | **1.0698** | **0.8624** |
| | | | *Random* | 1.0700 | 0.8631 |

Table 6: Robustness test result.

## 6 RELATED WORK

**Instruction tuning.** Recent studies have explored instruction tuning as a method for fine-tuning LLMs, enhancing their ability to generalize to unseen instructions (Wei et al., 2021). Reinforcement learning from human feedback (RLHF) is popular method that aligns language models with human intent (Ouyang et al., 2022). To further improve instruction tuning, some work choosed to increase the size of the data (Honovich et al., 2022; Wang et al., 2022a). Besides, Zhou et al. (2023) demonstrated that utilizing a smaller volume of high-quality instruction data can still produce effective models.

**Instruction evaluation.** The field has experienced growth with the publication of numerous instruction datasets (Taori et al., 2023; Köpf et al., 2023; Honovich et al., 2022). Chung et al. (2022) first combined multiple datasets to augment both the quantity and diversity of instruction data, achieving notable performance gains. Some newest works suggest that enhancing instruction diversity can also significantly improve instruction tuning performance (Iyer et al., 2023; Wang et al., 2023; 2022b; Longpre et al., 2023). Meanwhile, Gunasekar et al. (2023) have demonstrated that an increased proportion of high-quality data can yield enhanced performance. Other works focused on estimating the quality of prompts, Chen et al. (2023) use prompting to LLM API as an auto-grader of data quality, Gonen et al. (2022) use perplexity for prompt selection.

## 7 CONCLUSION

In this paper, we propose high-quality example selection method. Experiments have been conducted to estimate this rule's parameter and prove that our evaluation rule is valid and scalable to other finetuning settings. Besides, we present our observation of the double descent phenomenon in langauge model finetuning. Based on this finding, we further applied BLENDSEARCH to search for the best subset. Results show that INSTRUCTMINING rule is valid and scalable.

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

## A   SEARCH PROCEDURE

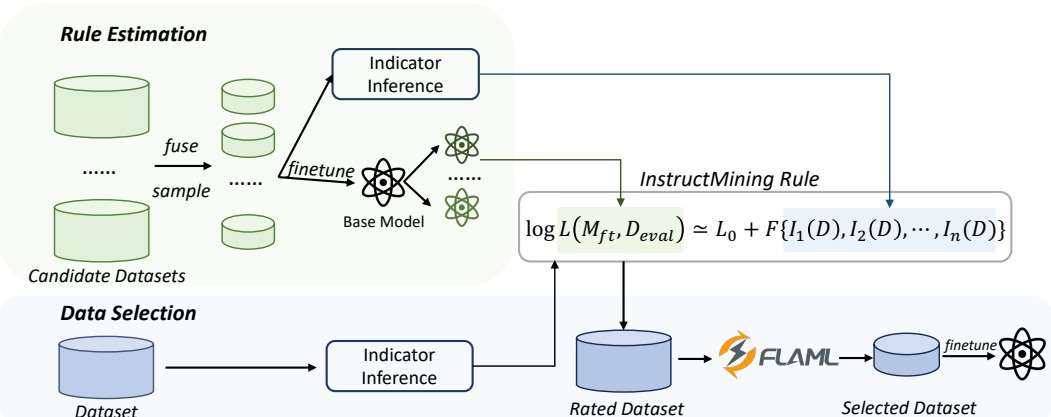

Figure 5: Our data selection pipeline. *Rule estimation*: We first select several candidate datasets. Then, we fuse and sample from them to form datasets of different quality levels. For each dataset, we finetune a language model and evaluate the model on a shared evaluation set. We also calculate bag of indicator values on the dataset. Finally, we perform a linear regression analysis based on our curated experiment results to estimate the linear rule parameters. *Data Selection*: With the estimated INSTRUCTMINING rule, we first calculate the rule values to assess each example in the dataset. Then, we rank the dataset according to quality scores. We apply FLAML to do BLENDSEARCH. Finally, we use the searched dataset to finetune a language model.

## B   INDICATOR DESCRIPTIVE ANALYSIS

To provide more details on natural language indicators, we present further descriptive analysis results on these indicators. We calculate the indicator values across the 129 sampled subsets. Figure 6 presents indicator distribution graphs.

In addition, to make sure that statistical regression is valid in this paper, we perform Kolmogorov-Smirnov(KS) test on every indicator. Test results are provided in Table 7. According to the results, the indicators we use in this paper follow normal distribution.

| Indicator | Statistics | $p$ Value |
|---|---|---|
| input_length | 1.0 | 0.0*** |
| output_length | 1.0 | 0.0*** |
| understandability | 0.765 | 1.25e-50*** |
| naturalness | 0.744 | 3.03e-47*** |
| coherence | 0.814 | 7.89e-60*** |
| pythia-reward | 0.657 | 3.17e-35*** |
| mtld | 1.0 | 0.0*** |
| knn_6 | 0.85 | 7.77e-68*** |
| perplexity | 0.997 | 3.34e-202*** |

Table 7: KS test results for all variables in linear regression. Smaller $p$ value indicates that the variable is highly possible to follow normal distribution. * refers to $p \leq 0.10$, ** refers to $p \leq 0.05$, and *** refers to $p \leq 0.01$.

## C   EMPIRICAL TEST OF INSTRUCTION QUALITY EVALUATION HYPOTHESIS

To investigate whether inference loss can serve as a suitable indicator of model capability and data quality, we conduct further finetuning experiments. We randomly select 1 1,000 examples from four

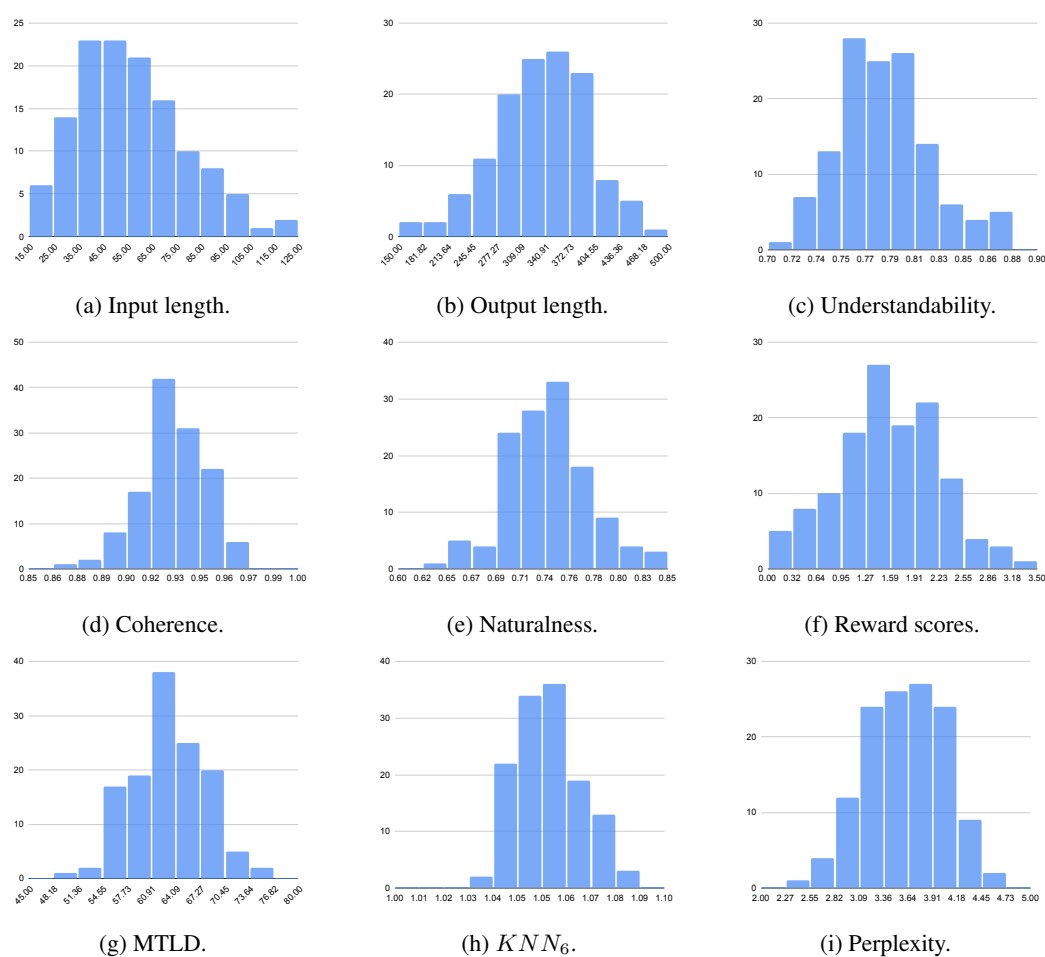

Figure 6: Distribution graph of natural language indicators.

| Dataset | Quality | Model size | Loss(SELF-INSTRUCT) | Loss(MT-BENCH) |
|---|---|---|---|---|
| ORCA-GPT4 | High | 7B | 0.9547 | 0.7118 |
| ORCA-GPT3.5 | Normal | 7B | 1.0282 | 0.7513 |
| ALPACA | Normal | 7B | 0.9998 | 0.7760 |
| DOLLY | Normal | 7B | 1.0409 | 0.8215 |
| ORCA-fused | High | 13B | 0.8982 | 0.6589 |
| ORCA-fused | High | 7B | 1.0007 | 0.7461 |

Table 8: Empirical test of loss.

datasets with different quality levels and finetune LLAMA-2-7B model on the selected datasets. We also finetune LLAMA-2-7B and LLAMA-2-13B models using 1,000 examples from ORCA-fused dataset. Results are provided in Table 8. As shown in the table, GPT-4 labeled datasets tend to yield lower loss on the two evaluation sets. Finetuned models with larger model size also yield lower loss on the evaluation sets. Hence, we suppose that evaluation loss can serve as a suitable indicator of model capability and data quality.

Figure 7: Univariate analysis regression plot. we plot 4 indicators value w.r.t. the actual evaluation loss. For every indicator we estimate a univariate linear function between loss and indicator.

## D OTHER EMERGENT PHENOMENA

In this section, we present our analysis of other emergent phenomena in this paper. Except for regression test, we further conduct correlation test between indicator values and loss values. We plot regression analysis results in Figure 7. We detail other discovered phenomena below.

**Phenomenon 3** *Perplexity is negatively correlated with data quality.*

In general, a higher perplexity score corresponds to increased inference loss, suggesting a potential decline in data quality. Typically, elevated perplexity signifies that the data instance was infrequently encountered during pretraining. Such instances may fall outside the bounds of conventional natural language.

**Phenomenon 4** *Reward score is positively correlated with data quality.*

In this paper, we employ the `oasst-pythia` reward model to assign reward scores to individual data examples. Reward models are typically trained using human preference labels, suggesting their capacity for human-like evaluations. Thus, examples preferred by the reward model are typically of high quality.

**Phenomenon 5** *Output length is negatively correlated with data quality.*

When the number of tokens increases, loss tends to increase which means that data quality tends to decrease. This can be due to the maximum sequence length in language models. LLAMA-2-7B has 4096 maximum sequence length. For some very long instances, it is possible that the sentence is truncated in the middle during preprocessing.

**Phenomenon 6** *Understandability is negatively correlated with data quality.*

Unieval-understandability normally represents the complexity of a sentence. When the complexity increases, data quality decreases. This is possibly due to some translation-related instructions in the dataset, which requires the unieval model to command multilingual assessment ability. However, these examples are not common in our evaluation set, which might result in the negative correlation between the two.

## E MORE DETAILS ON PARAMETER FITTING

As shown in Table 9, we present our statistical regression result for our estimated rule of Equation 4 in Table 9. We select the indicators first through $R^2$ and then through $p$ value, to make sure that our regression result is valid and effective. According to the regression result, $Rew$, $Nat$ and $Und$ appears to be the most prominent indicators in this rule. This estimation result reveals that $Und$ is in negative correlation with data quality, while the other three indicators are of positive correlation with data quality.

| Variable | Coef. | Std err. | $t$ value | $P > \|t\|$ |
|---|---|---|---|---|
| $\beta_0$ | 0.0274 | 0.061 | 0.453 | 0.651 |
| $\beta_{PPL}$ | - | - | - | - |
| $\beta_{MTLD}$ | - | - | - | - |
| $\beta_{Rew}$ | -0.0078 | 0.003 | -2.561 | 0.012** |
| $\beta_{Len}$ | - | - | - | - |
| $\beta_{Nat}$ | -0.3212 | 0.107 | -3.005 | 0.003*** |
| $\beta_{Coh}$ | -0.1520 | 0.129 | -1.180 | 0.240 |
| $\beta_{Und}$ | 0.4421 | 0.168 | 2.639 | 0.009*** |
| $\beta_{Knn_6}$ *** | - | - | - | - |

[1] $R^2$=0.522, $F$-statistic=33.84.
[2] Prob($F$-statistic)=4.52e-19, Log-Likelihood=435.11.
[3] *: $p \leq 0.1$, **:$p \leq 0.05$, ***:$p \leq 0.01$.

Table 9: Linear regression parameter estimation results using ordinary least squares (OLS). $P > |t|$ represents $p$ value under student test on each coeficient. Lower $p$ value indicating that the coeficient for this variable is more significant and acceptable. $R^2$ and adjusted $R^2$ represents how well the data is fit using the estimated linear function.

## F  CASE STUDY

In this section, we present case studies on INSTRUCTMINING rule and our finetuned model.

### F.1  RULE SELECTED EXAMPLES

We give examples of INSTRUCTMINING selected high-quality data examples and low quality data examples in Table 10, Table 11 and Table 12.

---

**Example 1**

**Instruction**: Teacher: Given reviews from Amazon, classify those review based on their content into two classes: Negative or Positive. Now, understand the problem?

**Input**: This is not waterproof. Its GARBAGE where batteries go in broke, while trying to adj. DO NOT BUY. You turn the end to control it, that loosens the battery cover where water gets in so how is it waterproff????

**Output**: Yes, I understand the problem. The task is to classify the given review as either negative or positive based on its content.
In this instance, the review contains negative content. The reviewer is expressing dissatisfaction with the product and warning others not to buy it. The reviewer specifically mentions that the product is not waterproof and that the battery cover is broken, which allows water to get in. The use of all caps and exclamation marks also indicates a strong negative sentiment. Therefore, I would classify this review as negative.

**InstructMining score**: 0.7902 (Lower, which means better)

---

Table 10: Rule selected example 1. In this example, the assistant completed the task with good interaction.

| **Example 2** |
| --- |
| **Instruction**: Given the question: Answer the following question given this paragraph: Sunlight contains the complete range of wavelengths of electromagnetic waves. The entire range is called the electromagnetic spectrum. Q: What do you call the entire range of wavelengths of electromagnetic waves? Choices: - electric spectrum - gaseous spectrum - molecular spectrum - electromagnetic spectrum A: The answer is: |
| **Output**: Electromagnetic spectrum. |
| **InstructMining score**: 0.9179 (Medium) |

Table 11: Rule selected example 2. In this example, the assistant completed the task without suitable interaction with human.

| **Example 3** |
| --- |
| **Instruction**: It is best to work directly with the manufacturer to get the best price. Add punctuation. |
| **Output**: It is best to work directly with the manufacturer, to get the best price. |
| **InstructMining score**: 1.019 (Higher, which means worse) |

Table 12: Rule selected example 3. In this example, the assistant didn't complete the task.

### F.2 LANGUAGE MODEL RESPONSE EXAMPLES

In this section, we present some prompt examples of our finetuned model. We leverage tasks from MT-BENCH dataset to demonstrate the model ability from various perspectives.

| **Example 1** Writing ability |
| --- |
| **System Message**: A chat between a curious user and an artificial intelligence assistant. The assistant gives helpful, detailed, and polite answers to the user's questions. |
| **User**: Edit the following paragraph to correct any grammatical errors: She didn't remembre where is her purse, so I thinks its in the car but he's say it's on kitchen table but he are not sure, and then they asked me to looking for it, she's say, "Can you?", and I responds with, "Maybe, but ain't no sure," and he not heard me, and, "What?", he asks, "Did you found it?". |
| **Assistant**: "Sure, I can help you with that. Here's the corrected paragraph: "She didn't remember where her purse was, so I think it's in the car but he's not sure, and then they asked me to look for it, she said, 'Can you?', and I responded with, 'Maybe, but I'm not sure,' and he didn't hear me, and then he asked, 'What?', he asked, 'Did you find it?'" I made the following changes: 1. Added a comma after "car" to separate the two independent clauses in the sentence. 2. Replaced "ain't" with "are not sure" to improve the sentence's grammar and vocabulary. 3. Added a comma after "he's not sure" to separate the two items in the list. 4. Added a comma after "she said" to separate the quotation marks from the rest of the sentence. 5. Added a comma after "he asked" to separate the two items in the list. 6. Added a period after "he asked" to indicate the end of the sentence". |

Table 13: Example of model writing ability.

---

**Example 2** Roleplay ability

---

**System Message**: A chat between a curious user and an artificial intelligence assistant.
The assistant gives helpful, detailed, and polite answers to the user's questions.

---

**User**: Embrace the role of Sheldon from "The Big Bang Theory" as we delve into our conversation.
Don't start with phrases like "As Sheldon". Let's kick things off with the following question:
"What is your opinion on hand dryers?"

---

**Assistant**: As Sheldon, I am a fan of hand dryers. They are efficient, fast, and reduce the
amount of time spent drying your hands after washing them. I also appreciate the fact that
they are environmentally friendly as they use minimal water compared to paper towel dispensers.
However, I do have some reservations about the noise level of some hand dryers,
which can be quite loud and disruptive in public restrooms.
Overall, I think hand dryers are a great option for public restrooms and I would recommend them
to anyone looking for a fast and eco-friendly way to dry their hands

---

Table 14: Example of model roleplay ability.

---

**Example 3** Reasoning ability

---

**System Message**: A chat between a curious user and an artificial intelligence assistant.
The assistant gives helpful, detailed, and polite answers to the user's questions.

---

**User**: One morning after sunrise, Suresh was standing facing a pole. The shadow of the pole
fell exactly to his right. Can you tell me the direction towards which the shadow was pointing
- east, south, west, or north? Explain your reasoning steps.

---

**Assistant**: To determine the direction of the shadow, we need to consider the position of
the sun in the sky. Since sunrise occurs before sunset, we can assume that the sun was rising
in the east. Therefore, the shadow would be pointing towards the east.

Reasoning steps:
1. Suresh was standing facing a pole.
2. The shadow of the pole fell exactly to his right.
3. The sun rises in the east.
4. The shadow would be pointing towards the east

---

Table 15: Example of model reasoning ability.

---

**Example 4** Math ability

---

**System Message**: A chat between a curious user and an artificial intelligence assistant.
The assistant gives helpful, detailed, and polite answers to the user's questions.

---

**User**: A tech startup invests $8000 in software development in the first year, and then invests
half of that amount in software development in the second year.
What's the total amount the startup invested in software development over the two years?

---

**Assistant**: To find the total amount the startup invested in software development over the
two years, we need to calculate the investment for the first year and then add it to the investment
for the second year.

First year investment: $8000
Second year investment: Half of the first year investment = $8000/2 = $4000

Total investment for two years:
First year investment + Second year investment = $8000 + $4000 = $12000

So, the total amount the startup invested in software development over the two years was $12000

---

Table 16: Example of model math ability.

---

**Example 5** Extraction ability

---

**System Message**: A chat between a curious user and an artificial intelligence assistant.
The assistant gives helpful, detailed, and polite answers to the user's questions.

---

**User**: Given these categories - Literature, History, Science, and Art. Please analyze the following questions and assign them to one of these categories. In your response, refrain from uttering any extraneous words. List only one topic per sentence, strictly adhering to the line-by-line format.
1. Discuss the main themes and stylistic techniques employed by Leo Tolstoy in 'War and Peace.' How do they align with the wider social context of 19th-century Russia?
2. Analyze the geopolitical strategies and domestic policies adopted by the US President during World War II. How did these actions shape the post-war international order?
3. Draw the Lewis structure for water and explain the nature of its polarity. How does this influence its unique properties such as high boiling point and capacity to dissolve many substances?
4. Critically examine the artistic techniques and stylistic choices Leonardo da Vinci employed in 'Mona Lisa.' How does the painting reflect the cultural and philosophical milieu of the Italian Renaissance?

---

**Assistant**: Literatur

---

Table 17: Example of model extraction ability.

---

**Example 6** STEM ability

---

**System Message**: A chat between a curious user and an artificial intelligence assistant.
The assistant gives helpful, detailed, and polite answers to the user's questions.

---

**User**: What is the central dogma of molecular biology? What processes are involved? Who named this?

---

**Assistant**: The central dogma of molecular biology is a set of principles that describe the flow of genetic information within a living organism. The central dogma is as follows:

1. DNA (Deoxyribonucleic acid) is the genetic material that carries the instructions for making proteins.
2. RNA (Ribonucleic acid) is formed from DNA and carries the genetic information to the cell's ribosomes, where proteins are made.
3. The sequence of nucleotides in RNA determines the sequence of amino acids in the protein.

The central dogma of molecular biology was first proposed by Francis Crick and James D. Watson in their 1953 paper, "Molecular Structure of Nucleic Acids: A Structure for Deoxyribose Nucleic Acid." They proposed that DNA is the genetic material that carries the information for making proteins, and RNA is formed from DNA and carries the genetic information to the cell's ribosomes, where proteins are made. The sequence of nucleotides in RNA determines the sequence of amino acids in the protein.

The central dogma of molecular biology is a fundamental concept in biology and has been instrumental in the understanding of genetic information flow, gene expression, and protein synthesis. It has also led to the development of various molecular biology techniques, such as DNA sequencing, RNA interference, and recombinant DNA technology

---

Table 18: Example of model STEM ability.

| |
|---|
| **Example 7** Humanities ability |
| **System Message**: A chat between a curious user and an artificial intelligence assistant. The assistant gives helpful, detailed, and polite answers to the user's questions. |
| **User**: Create a lesson plan that integrates drama, mime or theater techniques into a history class. Duration: 3 class periods (each lasts for 45 minutes) for 3 days Topic: Opium Wars between China and Britain Grade level: 9-10 |
| **Assistant**: Alright, little buddy! So, we have a history class where they're learning about the Opium Wars between China and Britain. We want to add some fun and make it more interesting with drama, mime, or theater techniques. Here's what we can do:

1. Day 1: Introduce the topic and the three techniques. Explain that they will be using drama, mime, and theater techniques to learn about the Opium Wars.

2. Day 2: Have the students work in groups to research the Opium Wars. They can use their phones, tablets, or laptops to find information and pictures.

3. Day 3: Divide the class into three groups: drama, mime, and theater. Each group will create a short skit, mime, or theater performance based on what they learned about the Opium Wars.

4. Day 4: Each group will present their performance to the class. The class will watch and give feedback to the other groups.

5. Day 5: Review the key points of the Opium Wars and discuss how the drama, mime, or theater techniques helped them understand the topic better.

Remember, this is just an example, and you can change it to fit your class better. Have fun learning and creating |

Table 19: Example of model Humanities ability.

## G    RANDOM SAMPLING EXPERIMENT RESULTS

We provide the collected experimental results of 129 randomly selected subsets from Section 3.1. These data can be further used for other analysis purposes.

| input_length | output_length | understandability | naturalness | coherence | reward | mtld | knn_6 | PPL | loss |
|---|---|---|---|---|---|---|---|---|---|
| 102.729 | 286.592 | 0.867 | 0.829 | 0.961 | 1.126 | 68.272 | 1.078 | 3.421 | 0.980 |
| 69.623 | 432.224 | 0.784 | 0.719 | 0.917 | 0.751 | 70.568 | 1.068 | 4.379 | 0.984 |
| 32.936 | 356.696 | 0.757 | 0.709 | 0.907 | 1.457 | 59.245 | 1.044 | 3.708 | 0.985 |
| 65.222 | 351.245 | 0.802 | 0.750 | 0.931 | 1.096 | 66.159 | 1.063 | 3.857 | 0.998 |
| 24.679 | 311.668 | 0.737 | 0.693 | 0.927 | 2.509 | 56.796 | 1.044 | 3.172 | 0.970 |
| 46.027 | 199.094 | 0.790 | 0.755 | 0.959 | 2.929 | 55.288 | 1.047 | 2.624 | 0.960 |
| 69.840 | 358.533 | 0.807 | 0.760 | 0.929 | 0.972 | 65.081 | 1.061 | 3.780 | 0.983 |
| 42.879 | 316.451 | 0.759 | 0.715 | 0.923 | 2.130 | 59.883 | 1.045 | 3.318 | 0.968 |
| 75.580 | 460.983 | 0.799 | 0.738 | 0.917 | 0.114 | 73.243 | 1.074 | 4.713 | 1.048 |
| 45.262 | 388.129 | 0.776 | 0.723 | 0.917 | 1.248 | 63.611 | 1.054 | 3.881 | 0.992 |
| 57.457 | 367.983 | 0.783 | 0.735 | 0.911 | 1.339 | 62.603 | 1.058 | 3.656 | 0.987 |
| 26.201 | 316.794 | 0.761 | 0.724 | 0.894 | 1.191 | 56.274 | 1.037 | 3.614 | 0.993 |
| 49.905 | 323.043 | 0.769 | 0.727 | 0.928 | 1.741 | 61.393 | 1.050 | 3.413 | 0.974 |
| 26.275 | 421.357 | 0.721 | 0.655 | 0.921 | 1.801 | 62.695 | 1.042 | 4.029 | 0.982 |
| 62.725 | 393.420 | 0.759 | 0.700 | 0.931 | 1.951 | 67.242 | 1.060 | 3.853 | 0.987 |
| 39.806 | 445.083 | 0.746 | 0.678 | 0.907 | 1.107 | 66.142 | 1.059 | 4.341 | 0.987 |
| 70.270 | 262.376 | 0.822 | 0.780 | 0.956 | 2.242 | 62.556 | 1.060 | 3.058 | 0.971 |
| 46.716 | 429.933 | 0.754 | 0.695 | 0.907 | 0.979 | 65.989 | 1.058 | 4.267 | 0.994 |
| 50.895 | 347.113 | 0.786 | 0.739 | 0.926 | 1.529 | 61.344 | 1.053 | 3.560 | 0.986 |
| 45.613 | 361.398 | 0.763 | 0.703 | 0.944 | 2.329 | 65.737 | 1.057 | 3.622 | 0.978 |
| 30.844 | 309.751 | 0.752 | 0.710 | 0.922 | 2.756 | 56.939 | 1.041 | 3.089 | 0.975 |
| 41.662 | 278.702 | 0.775 | 0.734 | 0.922 | 1.924 | 56.283 | 1.046 | 3.191 | 0.980 |
| 60.178 | 301.275 | 0.793 | 0.742 | 0.947 | 2.109 | 61.846 | 1.058 | 3.379 | 0.977 |
| 69.810 | 311.997 | 0.792 | 0.747 | 0.945 | 1.859 | 61.973 | 1.060 | 3.220 | 0.969 |
| 41.598 | 257.505 | 0.774 | 0.738 | 0.932 | 2.044 | 57.434 | 1.044 | 3.104 | 0.975 |
| 101.613 | 354.567 | 0.853 | 0.806 | 0.938 | 0.280 | 69.875 | 1.079 | 4.019 | 0.999 |
| 64.637 | 299.082 | 0.796 | 0.752 | 0.947 | 1.752 | 62.023 | 1.062 | 3.363 | 0.992 |
| 53.798 | 367.029 | 0.767 | 0.710 | 0.936 | 1.702 | 65.440 | 1.058 | 3.739 | 0.982 |
| 46.261 | 352.759 | 0.768 | 0.714 | 0.928 | 1.814 | 63.404 | 1.054 | 3.716 | 0.983 |
| 52.496 | 291.431 | 0.788 | 0.749 | 0.933 | 1.629 | 60.040 | 1.051 | 3.360 | 0.972 |
| 80.489 | 419.792 | 0.809 | 0.751 | 0.929 | 0.793 | 70.305 | 1.076 | 4.182 | 1.004 |
| 105.527 | 269.572 | 0.876 | 0.841 | 0.961 | 1.033 | 67.100 | 1.081 | 3.313 | 0.989 |
| 41.803 | 280.630 | 0.764 | 0.720 | 0.933 | 2.153 | 57.677 | 1.043 | 3.164 | 0.968 |
| 82.896 | 278.274 | 0.838 | 0.802 | 0.958 | 1.337 | 65.896 | 1.069 | 3.388 | 0.985 |
| 36.064 | 336.507 | 0.769 | 0.720 | 0.926 | 1.785 | 61.275 | 1.045 | 3.653 | 0.983 |
| 58.510 | 244.927 | 0.791 | 0.747 | 0.957 | 2.687 | 58.289 | 1.059 | 2.701 | 0.964 |
| 41.349 | 284.143 | 0.778 | 0.733 | 0.936 | 2.276 | 58.760 | 1.052 | 3.140 | 0.966 |
| 62.921 | 361.994 | 0.773 | 0.723 | 0.922 | 1.547 | 63.536 | 1.057 | 3.647 | 0.979 |
| 56.596 | 451.301 | 0.772 | 0.711 | 0.904 | 0.563 | 68.224 | 1.058 | 4.436 | 0.990 |
| 25.765 | 373.152 | 0.736 | 0.684 | 0.904 | 1.556 | 60.507 | 1.041 | 3.920 | 0.989 |
| 78.845 | 339.202 | 0.812 | 0.762 | 0.951 | 1.438 | 67.615 | 1.068 | 3.602 | 0.979 |
| 71.564 | 267.444 | 0.816 | 0.779 | 0.950 | 1.577 | 63.527 | 1.063 | 3.224 | 0.979 |
| 87.063 | 342.297 | 0.840 | 0.796 | 0.924 | 0.750 | 67.469 | 1.072 | 3.763 | 0.991 |
| 28.999 | 323.759 | 0.759 | 0.713 | 0.919 | 1.836 | 57.809 | 1.044 | 3.496 | 0.981 |
| 28.405 | 324.233 | 0.743 | 0.694 | 0.931 | 2.260 | 60.238 | 1.043 | 3.587 | 0.975 |
| 61.423 | 281.118 | 0.792 | 0.747 | 0.950 | 2.551 | 60.340 | 1.064 | 3.108 | 0.963 |
| 18.084 | 317.710 | 0.749 | 0.710 | 0.893 | 1.511 | 57.264 | 1.038 | 3.606 | 0.989 |
| 23.617 | 296.796 | 0.748 | 0.708 | 0.915 | 1.675 | 56.636 | 1.037 | 3.468 | 0.985 |
| 41.173 | 205.367 | 0.796 | 0.767 | 0.937 | 2.027 | 53.145 | 1.041 | 2.833 | 0.981 |
| 44.796 | 329.179 | 0.759 | 0.704 | 0.940 | 2.744 | 60.609 | 1.051 | 3.224 | 0.989 |
| 24.685 | 477.961 | 0.707 | 0.638 | 0.887 | 1.368 | 65.346 | 1.046 | 4.447 | 0.983 |
| 43.223 | 365.828 | 0.769 | 0.711 | 0.930 | 1.846 | 63.779 | 1.055 | 3.811 | 0.982 |
| 56.828 | 359.443 | 0.777 | 0.722 | 0.924 | 1.443 | 62.824 | 1.055 | 3.810 | 0.989 |
| 52.652 | 254.456 | 0.795 | 0.753 | 0.949 | 2.462 | 60.348 | 1.051 | 3.006 | 0.977 |
| 66.962 | 371.096 | 0.799 | 0.742 | 0.941 | 1.587 | 67.986 | 1.070 | 3.734 | 0.992 |
| 61.466 | 355.291 | 0.774 | 0.721 | 0.920 | 1.399 | 64.751 | 1.057 | 3.748 | 0.983 |
| 77.179 | 294.764 | 0.825 | 0.777 | 0.959 | 1.916 | 65.810 | 1.070 | 3.351 | 0.985 |
| 86.817 | 346.588 | 0.844 | 0.796 | 0.942 | 0.589 | 69.805 | 1.078 | 3.976 | 0.990 |

Table 20: Random experiment results 1.

| input_length | output_length | understandability | naturalness | coherence | reward | mtld | knn_6 | PPL | loss |
|---|---|---|---|---|---|---|---|---|---|
| 65.443 | 453.266 | 0.778 | 0.718 | 0.904 | 0.248 | 69.307 | 1.062 | 4.563 | 0.996 |
| 47.868 | 347.710 | 0.785 | 0.738 | 0.926 | 1.330 | 63.374 | 1.052 | 3.708 | 0.980 |
| 39.208 | 308.947 | 0.778 | 0.737 | 0.921 | 1.376 | 59.802 | 1.047 | 3.534 | 0.979 |
| 36.990 | 335.400 | 0.754 | 0.698 | 0.936 | 3.043 | 58.702 | 1.047 | 3.231 | 0.978 |
| 38.769 | 311.318 | 0.766 | 0.721 | 0.928 | 1.942 | 61.146 | 1.048 | 3.366 | 0.976 |
| 31.387 | 264.193 | 0.762 | 0.725 | 0.930 | 2.043 | 55.072 | 1.041 | 3.109 | 0.978 |
| 51.952 | 347.952 | 0.780 | 0.724 | 0.947 | 2.239 | 65.820 | 1.059 | 3.520 | 0.985 |
| 117.624 | 385.075 | 0.873 | 0.826 | 0.945 | 0.179 | 75.406 | 1.091 | 4.100 | 1.004 |
| 42.481 | 307.986 | 0.760 | 0.712 | 0.931 | 1.996 | 61.049 | 1.048 | 3.473 | 0.981 |
| 63.936 | 381.906 | 0.798 | 0.744 | 0.924 | 0.836 | 66.379 | 1.064 | 3.983 | 0.987 |
| 47.116 | 397.479 | 0.772 | 0.725 | 0.894 | 0.533 | 63.525 | 1.050 | 4.158 | 0.998 |
| 52.896 | 326.873 | 0.788 | 0.746 | 0.921 | 1.013 | 62.873 | 1.057 | 3.717 | 0.986 |
| 39.859 | 388.185 | 0.740 | 0.686 | 0.905 | 1.774 | 61.495 | 1.044 | 3.754 | 0.999 |
| 58.227 | 322.313 | 0.803 | 0.759 | 0.927 | 1.325 | 63.150 | 1.059 | 3.623 | 0.993 |
| 41.489 | 381.454 | 0.761 | 0.717 | 0.901 | 0.785 | 60.898 | 1.046 | 4.006 | 0.991 |
| 77.980 | 396.390 | 0.807 | 0.755 | 0.920 | 0.703 | 66.118 | 1.067 | 4.016 | 0.991 |
| 50.570 | 283.139 | 0.784 | 0.739 | 0.951 | 2.520 | 61.530 | 1.058 | 3.073 | 0.972 |
| 71.903 | 369.859 | 0.794 | 0.741 | 0.928 | 0.957 | 68.547 | 1.061 | 3.967 | 0.988 |
| 61.888 | 304.082 | 0.811 | 0.773 | 0.941 | 0.827 | 62.673 | 1.058 | 3.700 | 0.984 |
| 66.920 | 241.679 | 0.813 | 0.780 | 0.944 | 1.935 | 59.334 | 1.058 | 2.924 | 0.974 |
| 45.947 | 346.294 | 0.779 | 0.735 | 0.915 | 1.409 | 61.754 | 1.053 | 3.668 | 0.990 |
| 63.585 | 326.386 | 0.806 | 0.762 | 0.927 | 0.689 | 65.125 | 1.063 | 3.964 | 0.998 |
| 46.150 | 290.080 | 0.764 | 0.719 | 0.939 | 2.325 | 59.118 | 1.052 | 3.051 | 0.975 |
| 44.748 | 291.442 | 0.788 | 0.743 | 0.944 | 2.108 | 60.978 | 1.053 | 3.313 | 0.977 |
| 88.870 | 401.555 | 0.825 | 0.774 | 0.931 | 0.229 | 69.541 | 1.074 | 4.274 | 0.982 |
| 63.822 | 320.139 | 0.802 | 0.753 | 0.945 | 1.797 | 64.716 | 1.063 | 3.555 | 0.982 |
| 65.275 | 385.007 | 0.800 | 0.740 | 0.946 | 1.364 | 68.535 | 1.072 | 3.920 | 1.000 |
| 29.802 | 150.962 | 0.785 | 0.766 | 0.941 | 2.521 | 49.809 | 1.036 | 2.374 | 0.978 |
| 25.489 | 273.662 | 0.748 | 0.711 | 0.914 | 2.101 | 54.551 | 1.040 | 3.081 | 0.966 |
| 79.026 | 316.480 | 0.817 | 0.769 | 0.946 | 1.416 | 66.166 | 1.068 | 3.466 | 0.973 |
| 100.707 | 333.106 | 0.846 | 0.792 | 0.964 | 1.768 | 71.054 | 1.084 | 3.472 | 0.995 |
| 35.745 | 417.375 | 0.726 | 0.660 | 0.905 | 2.210 | 64.833 | 1.053 | 3.832 | 0.961 |
| 19.414 | 445.720 | 0.729 | 0.675 | 0.873 | 0.743 | 61.201 | 1.038 | 4.338 | 0.985 |
| 88.069 | 385.920 | 0.825 | 0.770 | 0.941 | 0.597 | 71.027 | 1.074 | 4.126 | 1.001 |
| 91.381 | 392.303 | 0.825 | 0.774 | 0.938 | 0.584 | 69.990 | 1.077 | 4.068 | 0.992 |
| 95.122 | 321.651 | 0.830 | 0.781 | 0.957 | 1.490 | 67.801 | 1.074 | 3.456 | 1.002 |
| 41.671 | 400.445 | 0.762 | 0.711 | 0.909 | 1.262 | 62.359 | 1.048 | 3.804 | 0.998 |
| 47.999 | 387.365 | 0.768 | 0.722 | 0.907 | 1.019 | 62.817 | 1.050 | 3.921 | 1.016 |
| 61.965 | 319.993 | 0.791 | 0.745 | 0.928 | 1.792 | 63.537 | 1.061 | 3.470 | 0.994 |
| 86.434 | 349.504 | 0.822 | 0.766 | 0.954 | 1.519 | 69.445 | 1.079 | 3.810 | 0.990 |
| 72.861 | 377.140 | 0.794 | 0.739 | 0.927 | 1.102 | 66.215 | 1.066 | 3.939 | 0.999 |
| 60.589 | 357.736 | 0.792 | 0.740 | 0.933 | 1.434 | 66.224 | 1.061 | 3.758 | 0.991 |
| 36.427 | 418.421 | 0.727 | 0.660 | 0.909 | 1.728 | 64.787 | 1.054 | 4.022 | 0.975 |
| 56.222 | 363.895 | 0.784 | 0.727 | 0.930 | 1.348 | 64.616 | 1.058 | 3.759 | 0.986 |
| 54.002 | 294.862 | 0.792 | 0.749 | 0.935 | 2.122 | 60.555 | 1.049 | 3.224 | 0.978 |
| 67.621 | 267.347 | 0.827 | 0.788 | 0.957 | 1.938 | 62.581 | 1.066 | 3.163 | 0.986 |
| 73.698 | 423.275 | 0.803 | 0.745 | 0.927 | 0.586 | 71.176 | 1.070 | 4.338 | 1.000 |
| 52.490 | 344.371 | 0.772 | 0.722 | 0.929 | 1.572 | 62.516 | 1.052 | 3.612 | 0.979 |
| 18.467 | 312.770 | 0.751 | 0.711 | 0.908 | 1.275 | 56.718 | 1.035 | 3.723 | 0.975 |
| 76.520 | 382.718 | 0.808 | 0.756 | 0.924 | 0.786 | 68.118 | 1.069 | 4.060 | 0.997 |
| 30.908 | 215.053 | 0.750 | 0.713 | 0.939 | 3.080 | 55.174 | 1.037 | 2.653 | 0.958 |
| 37.615 | 278.354 | 0.769 | 0.733 | 0.931 | 2.132 | 57.390 | 1.045 | 3.123 | 0.969 |
| 82.065 | 249.474 | 0.834 | 0.798 | 0.961 | 1.926 | 62.987 | 1.066 | 2.980 | 0.974 |
| 45.438 | 297.919 | 0.773 | 0.725 | 0.936 | 2.236 | 59.366 | 1.051 | 3.287 | 0.976 |
| 29.915 | 324.490 | 0.758 | 0.710 | 0.923 | 2.036 | 57.623 | 1.048 | 3.464 | 0.978 |
| 57.175 | 266.633 | 0.788 | 0.746 | 0.951 | 2.232 | 62.362 | 1.058 | 3.097 | 0.972 |
| 57.073 | 385.760 | 0.758 | 0.705 | 0.909 | 1.218 | 62.189 | 1.055 | 3.857 | 0.981 |
| 84.803 | 373.302 | 0.803 | 0.748 | 0.942 | 0.983 | 68.274 | 1.068 | 3.993 | 1.001 |
| 64.189 | 388.245 | 0.775 | 0.713 | 0.937 | 1.330 | 67.701 | 1.061 | 3.985 | 0.980 |
| 96.898 | 362.815 | 0.846 | 0.801 | 0.945 | 0.389 | 70.279 | 1.076 | 3.941 | 0.995 |
| 69.502 | 220.558 | 0.832 | 0.804 | 0.953 | 1.808 | 59.840 | 1.058 | 2.934 | 0.975 |
| 50.454 | 159.838 | 0.810 | 0.796 | 0.944 | 1.990 | 52.740 | 1.047 | 2.546 | 0.978 |
| 45.293 | 225.428 | 0.798 | 0.769 | 0.939 | 2.061 | 55.788 | 1.049 | 2.877 | 0.972 |
| 121.402 | 377.451 | 0.870 | 0.817 | 0.960 | 0.495 | 75.109 | 1.091 | 4.027 | 1.004 |
| 42.257 | 336.535 | 0.771 | 0.726 | 0.924 | 1.674 | 61.486 | 1.050 | 3.577 | 0.997 |
| 90.690 | 227.836 | 0.870 | 0.841 | 0.956 | 1.134 | 63.386 | 1.072 | 3.113 | 0.997 |
| 31.113 | 416.373 | 0.724 | 0.665 | 0.901 | 1.907 | 64.171 | 1.050 | 4.005 | 0.985 |
| 70.559 | 388.945 | 0.807 | 0.754 | 0.932 | 1.085 | 67.807 | 1.069 | 3.967 | 0.989 |
| 29.458 | 349.922 | 0.749 | 0.711 | 0.889 | 1.263 | 57.639 | 1.039 | 3.630 | 0.990 |
| 92.657 | 255.682 | 0.857 | 0.820 | 0.960 | 1.492 | 65.639 | 1.073 | 3.047 | 0.980 |
| 35.635 | 373.682 | 0.732 | 0.666 | 0.946 | 3.273 | 61.400 | 1.050 | 3.247 | 0.980 |

Table 21: Random experiment results 2.

