# OpenReview forum: "Instruction Mining: Instruction Data Selection for Tuning Large Language Models"
_ICLR.cc/2024/Conference — Submitted to ICLR 2024_

### Official Review · Reviewer_CNR2 · 2023-10-29

**Soundness:** 2 fair
**Presentation:** 3 good
**Contribution:** 3 good
**Rating:** 6
**Confidence:** 4

**Summary:**

The authors propose a technique, InstructionMining, to analyse the performance of a set of instruction-tuned models to identify the features that best correlate with model performance (as measured by loss on a validation set). They also apply BlendSearch to find the optimal data size to select. They show that selecting data with InstructMining results in models that perform as well or better than models trained on random data and much larger datasets (e.g. StableBeluga). They also find that dataset size and model performance display double descent, where increasing data helps, then hurts, then helps again, counter to intuition (that more data is always better).

**Strengths:**

- The method is sound, and investigating how to determine the ‘quality’ of a dataset without training a model is interesting and impactful for the community. The exploration of different indicators (and ablating them in 4.3) is interesting and has some useful takeaways, suggesting that just examining the naturalness and understandability of responses is enough to detect quality data.
- The GPT -4-based preference results suggest that the model trained on the InstructMining selected data performs better than random selection, and is similar in quality to Vicuna 1.5 7B.
- The results investigating dataset size and model performance are interesting and provide interesting guidance for future efforts in collecting data, as well as (partially) validating the LIMA hypothesis that fewer, but higher-quality, samples are preferable to a large amount of data.

**Weaknesses:**

- The method is computed over loss, but it is unclear how this translates to eval task performance. Appendix C presents a somewhat weak argument - it would be good to e.g. strongly correlate the inference loss with MMLU/MT-Bench/etc results. It’s non-obvious that this is true in instruction tuning setups due to the lack of hard gold labels. There may be many potential outputs that are equally valid, and only one is generated as the gold label by GPT-4. This also makes it unclear to determine what differences in loss are significant - for example, in Table 3, Random 150(Train) gets .0085 higher loss than Selected 22(Rule) + 150(Train). Is this difference significant? Does it actually translate to downstream task gains?
- The lack of multiple random seeds/error bars with the experiments makes it unclear how well InstructMining is performing over randomly selected data. Looking at the results in Table 4 especially, it seems that randomly selecting data outperforms InstructMining-Selected at the 10k size, and only underperforms the Selected model by .7 at the 40k size, which might not be a significant difference. While it’s expensive to run multiple trials, especially for larger data sizes, I think it’s important to make sure that the claimed performance improvements are actually significant.
- Missing baselines: it would be good to compare against models trained on all data from the different base datasets: just open orca, just dolly, and the two combined. This would allow us to examine how the InstructMining models compare to models trained on all available data while keeping the finetuning setup consistent. Currently, the main baselines are Vicuna, Llama-chat, and StableBeluga, none of which were trained on OpenOrca or Dolly.
- This is minor, but the names used in Table 4 could be explained better. I’m assuming InstructMining-Random is randomly selecting over OpenOrca and Dolly.

Overall, I like the work and think the proposed method is interesting and insightful, but I worry that the results may not be as clear-cut as presented in the work, due to the limited evaluation setup (a focus on loss, and only minor differences in performance on downstream tasks in table 4).

**Questions:**

- How large/small of a difference in loss is needed to translate to significant performance differences in downstream tasks?
- Do you have results with multiple trials for the random selection? Does this change any findings e.g. in the double descent analysis?
- In section 5.2, are you using the instruction rule derived from the 7b model for the 13b data selection?
- How do the InstructMining-trained models compare to models just trained on either Dolly or OpenOrca (or both together)? StableBeluga’s finetuning set has not been publicly released at the time of writing (as far as I know), so these would be good baselines to have.

---

> ### Author Response · Authors · 2023-11-22
>
> We apologize for the delayed response. Please note that some of our experiments and evaluations require several days to complete, which may result in a brief wait time before we are able to provide a reply. Thank you for your patience and we appreciate your time and effort in reviewing our paper and providing valuable feedback. We are pleased to hear your positive remarks regarding the novelty and practical applicability of our method for selecting high-quality instruction following data. In response to your concerns, we have outlined our comments below:
>
> - **Q.1 Difference in loss:**
>
> It’s hard to say the numerical relevance between loss and performance difference, here is the OpenLLM leaderboard result for random and selected 10k data of dolly.
>
> |  Model Name  | Datasize | loss@self-instruct | loss@mt-bench | lm-eval-avg | ARC    | Hellaswag | MMLU   | TruthfulQA |
> | -------------- | -------- | ------------------ | ------------- | ----------- | ------ | --------- | ------ | ---------- |
> | dolly random   | 10000    | 1.0356             | 0.8086        | 0.5486      | 0.5733 | 0.8116    | 0.4266 | 0.3828     |
> | dolly Selected | 10000    | 1.0371             | 0.8001        | 0.5495      | 0.5682 | 0.8123    | 0.4392 | 0.3782     |
>
> However, we would like to point out that, during our experimentation, we feel that loss on mt-bench, also LLM-as-a-judge, is only able to evaluate a model’s conversational ability, while a language model still needs other abilities, e.g., question answering. And this is why we add OpenLLM benchmarks. From our point of view, Mt-bench and MMLU are two separate evaluation benchmarks, which does not correlate much. For example, a response below could yield high LLM-as-a-judge score and low loss, however, it is not a correct answer to this question[1].
>
> ```
> Instruction: Sort the following list into alphabetical order. apple, banana, orange, grape.
>
> Wrong Answer: No problem! Here’s the sorted list. Grape, apple, banana, orange.
>
> Correct Answer: apple, banana, grape, orange.
> ```
> - **Q.2 Results on random selection experiments.**
>
> Thanks for pointing out. We will update more random results. These results and analysis will be added to our paper.
> Below is the detail for the appended random experiment results.
>
> | Model Name | Data size | Loss@self-instruct | Loss@mt-bench |
> | ------------------- | ----- | ----- | ----- |
> | figure1 orca Random | 1000  | 1.001 | 0.746 |
> | orca Random         | 1000  | 0.974 | 0.715 |
> | orca Random         | 1000  | 0.970 | 0.715 |
> | figure1 orca Random | 2000  | 0.966 | 0.711 |
> | orca Random         | 2000  | 0.980 | 0.713 |
> | orca Random         | 2000  | 0.990 | 0.713 |
> | figure1 orca Random | 3000  | 0.993 | 0.727 |
> | orca Random         | 3000  | 1.013 | 0.728 |
> | orca Random         | 3000  | 1.000 | 0.736 |
> | figure1 orca Random | 4000  | 1.014 | 0.752 |
> | orca Random         | 4000  | 0.999 | 0.751 |
> | orca Random         | 4000  | 1.015 | 0.728 |
> | figure1 orca Random | 5000  | 0.991 | 0.751 |
> | orca Random         | 5000  | 0.999 | 0.738 |
> | orca Random         | 5000  | 0.998 | 0.734 |
> | table4 orca Random  | 10000 | 1.020 | 0.750 |
> | orca Random         | 10000 | 1.012 | 0.757 |
> | orca Random         | 10000 | 1.003 | 0.742 |
> | table4 orca Random  | 40000 | 1.015 | 0.734 |
> | orca Random         | 40000 | 1.019 | 0.732 |
> | orca Random         | 40000 | 1.011 | 0.754 |
>
> We summarize this table in to the analysis below:
>
> | Datasize | Metric                    | Average | Std.           | Value       |
> | -------- | ------------------------- | ------- | -------------- | ----------- |
> | 1000     | avg/std of loss           | 0.981   | 0.0169  | 0.981±0.017 |
> | 1000     | avg/std of loss(mt bench) | 0.725   | 0.0178  | 0.725±0.018 |
> | 2000     | avg/std of loss           | 0.979   | 0.0117  | 0.979±0.012 |
> | 2000     | avg/std of loss(mt bench) | 0.713   | 0.0014 | 0.713±0.001 |
> | 3000     | avg/std of loss           | 1.002   | 0.0104  | 1.002±0.01  |
> | 3000     | avg/std of loss(mt bench) | 0.731   | 0.0048  | 0.731±0.005 |
> | 4000     | avg/std of loss           | 1.009   | 0.0091 | 1.009±0.009 |
> | 4000     | avg/std of loss(mt bench) | 0.744   | 0.0135   | 0.744±0.014 |
> | 5000     | avg/std of loss           | 0.996   | 0.0043 | 0.996±0.004 |
> | 5000     | avg/std of loss(mt bench) | 0.741   | 0.0089 | 0.741±0.009 |
> | 10000    | avg/std of loss           | 1.012   | 0.0083 | 1.012±0.008 |
> | 10000    | avg/std of loss(mt bench) | 0.750   | 0.0072 | 0.75±0.007  |
> | 40000    | avg/std of loss           | 1.015   | 0.0039 | 1.015±0.004 |
> | 40000    | avg/std of loss(mt bench) | 0.740   | 0.0351  | 0.74±0.035  |
> We will also update the figure in our paper with this result.
>
> - **Q.3 Section 5.2, the usage of instruction rule.**
>
> Yes, we use data with rule estimated by 7B model to train 13B model.

---

> ### Author Response · Authors · 2023-11-22
>
> [Continued]
>
> - **W.3 & Q.4 Models trained on full dataset.**
>
> Orca full is too large for us to train at this moment, here we provide our updated result on full dolly(15k) dataset.
>
> |Model | Datasize | Loss@self-instruct | Loss@mt-bench | OpenLLM Avg| ARC| Hellaswag| MMLU|TruthfulQA|
> |---|---|---|---|---|---|---|---|---|
> |Dolly-full| 15,000| 1.047 | 0.807 | 0.546 | 0.566 | 0.808 | 0.437 | 0.371 |
> This result will also be added into our paper.
>
> - **W.1 The correlation between LLM-as-a-judge scores and loss.**
>
> In table4, we bring loss and LLM-Leaderboard together:
>
> | Model                   | Data size | Loss        | Loss(MT-BENCH) | Avg. Metric | ARC   | HellaSwag | MMLU  | TruthfulQA |
> | ----------------------- | --------- | ----------- | -------------- | ----------- | ----- | --------- | ----- | ---------- |
> | INSTRUCTMINING-Selected | 10,000    | 0.9767      | 0.7177         | 58.65       | 56.66 | 79.77     | 49.89 | 48.26      |
> | INSTRUCTMINING-Selected | 40,000    | 1.006       | 0.7462         | 59.25       | 54.44 | 80.11     | 52.6  | 49.83      |
> | INSTRUCTMINING-Random   | 10,000    | 1.012±0.008 | 0.75±0.007     | 58.74       | 54.78 | 79.58     | 49.02 | 51.58      |
> | INSTRUCTMINING-Random   | 40,000    | 1.015±0.004 | 0.74±0.035     | 58.95       | 54.78 | 79.89     | 51.16 | 49.95      |
>
> We would like to kindly point out that, since our optimization goal is to minimize the distance between the finetuned model and the target model, hence, GPT-4 in our paper, it is possible that MMLU does not correlate much with the loss. A straightforward intuition is that,  MMLU is more like a question answering task, while our loss minimization goal aims to improve the model’s ability in conversation. In this case, we choose to relate loss with mt-bench, which is also LLM-as-a-judge scores to prove that loss can serve as a suitable indicator for model performance.
>
> We are still downloading model weights and running experiments on 7B open-source models. We will provide the experiment results as soon as it is available.
>
> - **W.4 Name explanation.**
>
> Thanks for pointing out the ambiguity in our paper. In Table 4, both InstructionMining-Selected and InstructionMining-Random are over Orca.
>
> Thanks again for your constructive feedback. We feel truly encouraged that you like the idea and find it insightful.
>
> *Reference:*
>
> [1] Zeng, Zhiyuan, et al. "Evaluating large language models at evaluating instruction following." arXiv preprint arXiv:2310.07641 (2023).

---

> ### Comment · Reviewer_CNR2 · 2023-11-22
>
> Hi, thanks for your response and the additional experiments! Thank you for the clarifications, they are useful.
>
> - **Correlation between loss & downstream performance**:
> Thanks for these results! I guess your strong MT-Bench results in figure 3 suggest that optimising for loss works for improving 'real' MT-Bench results. It still would be nice to get some idea of how large of a +/- change in loss you need to see a difference in performance on MT-Bench. I guess this also suggests your derived rule is focusing on conversational ability, rather than other model capabilities - you would have to test if learning a new rule would work for improving e.g. multiple-choice ability (MMLU).
>
> - **Different seeds**:
> I was especially curious about the variance in results over downstream metrics, either the gpt-annotation-based version of MT-Bench or open-llm benchmark scores (although, as you noted, if your method works better for general conversation ability, then just examining mt-bench variation is fine).
>
> - **Full Dolly results**:
> Thank you for these - examining Table 3 in the paper, it seems like your method does generally better in mt-bench loss than training on the entire set, which is interesting!
>
> I have carefully read the other reviews and responses and am keeping my score.

---

### Official Review · Reviewer_TXJx · 2023-10-31

**Soundness:** 2 fair
**Presentation:** 3 good
**Contribution:** 2 fair
**Rating:** 3
**Confidence:** 4

**Summary:**

The paper proposed a method for instruction selection. The main idea is that the instruction quality of D can be estimated through the inference loss of Mf t on a evaluation dataset Deval. Experiment results show that InstructMining-7B performs well.

**Strengths:**

1. The paper is generally well-written and easy to follow.
2. pioneer the application of traditional data mining techniques to augment Language Learning Models (LLMs) by autonomously curating data of superior quality.

**Weaknesses:**

1. I think the method in this paper may not be novel and solid. Whether selecting another dataset as D-eval is a good choice is not sufficiently justified. The distributions of D-eval and the target task are different. The paper does not touch the core problem, i.e., how to select and estimate the effect of some instructions with a biased/smaller dataset.
2. The paper posits that the instruction quality of D can be estimated through the inference loss of M-ft on a evaluation dataset D-eval. However, the negative log-likelihood (NLL) reflects the extent to which a language model trained on D can explain the data in D-eval, essentially measuring the textual similarity between D and D-eval. If the quality of D-eval is not high enough, this method of assessing instruction quality could be biased. Furthermore, how can we ensure the quality of D-eval? Does a higher similarity to D-eval necessarily indicate a higher quality of D?
3. The paper associates the optimal point with the lowest loss, selecting the data that can achieve the lowest loss as the optimal data. However, it is important to note that there is no direct correlation between the training loss and the quality of data. An output with a longer length often results in a higher training loss, but this does not necessarily indicate lower quality. The paper does not take this factor into account.
4. Each time D-eval changes (for instance, when a higher quality D-eval is chosen), the multivariate linear function must be relearned. In other words, the method proposed in the paper lacks scalability.
5. It would be great to see experiments conducted using the same volume of data for model training on both LIMA and the INSTRUCTION MINING method proposed in the paper, followed by a comparison of their respective performances on MT-BENCH.

**Questions:**

Please refer to the weaknesses.

---

> ### Author Response · Authors · 2023-11-16
>
> We thank Reviewer TXJx for the thoughtful feedback. Your positive comments on our writing and our method for selecting data of superior quality is truly encouraging. For your concerns and questions for our work, please find response below.
>
> - **Q.1.1 Choice of D-eval.**
>
> Thanks for pointing this out. Indeed, we recognize that we didn’t explain in detail why we selected another D-eval, instead of just choosing an evaluation subset from the training data. When selecting D-eval, We want
> 1. D-eval to have high-quality instruction following examples in it.
> 2. D-eval to be diverse enough, hence, it should have:
> (a). various instructions to test the model’s generalizability;
> (b). High-quality responses.
> 3. D-eval to be relatively small, so that we can conduct the experiments efficiently.
>
> At the time of submission, we only found two suitable, human labeled instruction datasets, self-instruct and mt-bench. To make sure that our selected D-eval satisfies these requirements, we also re-annotated self-instruct dataset using GPT-4 to get high-quality responses.
>
> - **Q.1.2 Distribution difference between D-eval, target tasks, and D**
>
> We agree that if we directly apply this rule to select data of other target tasks, our rule will probably not work as expected. However, we would like to point out that our paper mainly focuses on instruction tuning, hence, teaching the model to follow instructions. The datasets we choose to train the model and fitting the rule and mostly instruction-following data, and we mainly focus on tuning the model to make it better at following instructions. In this case, our task is general instruction following, which aims to tune an LLM to follow instructions. So there should not be much difference in the distribution of D-eval and target tasks.
>
> - **Q.1.2 How to select and estimate the effect of instructions with a biased/smaller dataset.**
>
> We agree that if the D-eval is biased or too small, our method might not work under these circumstances. We want to kindly point out that our method is highly dependent on a suitable, high-quality D-eval dataset. A strong hypothesis in this paper is that D-eval is of relatively high quality. In this case, selecting and estimating instruction quality using a biased D-eval is out of our current scope. However, selecting data when D-eval is biased or too small is of course an important research topic, and hopefully in our future research, we can keep on improving our method to make it more robust, towards different D-eval sets.
>
> - **Q.2 The correlation between selected D and D-eval.**
>
> We acknowledge that using D-eval for fitting the rule and then using the rule to select data will possibly result in overfitting on the D-eval. This could harm performance if D-eval is of low-quality. However, if D-eval is of high-quality, the selected D should be of similar quality. Hence, our method is strongly dependent on the hypothesis that D-eval is of high-quality. To ensure its quality, which is the self-instruct dataset in our paper, we re-annotated self-instruct dataset using GPT-4, to ensure that the label of self-instruct is of high-quality.
>
> - **Q.3 Training loss, data quality and output length.**
>
> Instead of using training loss, we use model inference loss on D-eval, so D-eval is an unseen dataset to the model. We recognize that inference loss cannot accurately represent a generative model’s performance. However, inference loss is still an indicator of the textual similarity between responses generated by GPT-4 and the trained model, and it is able to indicate data quality to some extent, as shown in Table 8. Choosing a suitable metric to evaluate a model’s performance or data quality requires us to consider both representability and efficiency, which still remains an unsolved problem in the research area. Inference loss is like a “compromise” that it is able to represent a model’s ability in following instructions to some extent and also save inference time.
>
> Also, we would like to clarify that our analysis in appendix section D is only a descriptive analysis over the linear similarity. For example, output length in appendix D shows a positive linear correlation with loss, however, this does not mean that output length is strongly correlated with loss in other spaces. During rule fitting, under the combined influence of multiple indicators, output length becomes less significant, which results in its absence in the final estimated law.
>
> We also would like to note that, during experimentation, we found finding a suitable metric quite hard since a generative language model cannot be simply evaluated only from its conversation ability or from its question answering ability. [Recent works](https://arxiv.org/abs/2310.07641) also discover that even GPT-4 could be “tricked” by superficial expressions[1]. However, expanding our optimization objective to multiple metrics is out of our current scope.

---

> > ### Author Response · Authors · 2023-11-16
> >
> > *[Continued]*
> >
> >
> > - **Q.4 The expense of changing D-eval.**
> >
> > Normally, you don’t need to redo the experiments. If a D-eval with significantly higher quality is out, e.g. GPT-5 labeled D-eval, and we want a better law to select examples which are close to this D-eval set, then we need to redo the fitting experiments. However, the expense is still affordable. During our experimentation, every data collection experiment takes ~15mins to complete, and the total fitting experiments takes around 32 hours to complete.
> >
> > - **Q.5 Comparison between LIMA and Instruction Mining.**
> >
> > We would be able to provide our experimental results in a few days. We also promise that the comparison result would be added to our paper. Once the result is out, we will also add it into our response.
> >
> > *Reference*:
> >
> > [1] Zeng, Zhiyuan, et al. "Evaluating large language models at evaluating instruction following." arXiv preprint arXiv:2310.07641 (2023).

---

> ### Author Response · Authors · 2023-11-19
>
> Dear Reviewer TXJx,
>
> We just got the result of LIMA-7b model (LLaMA-2-7B finetuned with LIMA data). Its OpenLLM benchmark scores are shown in the table below.
> |Model | Datasize | loss@self-instruct| loss@mt-bench| OpenLLM Avg| ARC| Hellaswag| MMLU|TruthfulQA|
> |---|---|---|---|---|---|---|---|---|
> |LIMA-7b|1000|0.9947|0.7604|0.5533|0.5563|0.8009|0.4371|0.419|
> |InstructMining-7b|1000| 0.9576 | 0.7114 | 0.562525  | 0.5589 | 0.7877 | 0.4299 | 0.4736 |
>
> We also compared its performance with our finetuned InstructMining model using LLM-as-a-judge. Results are listed in the table below.
> | Model | Win Rate | Lose Rate | Tie Rate | Adjusted Win Rate |
> |---|---|---|---|---|
> |InstructMining-7b(vs LIMA-7b)| 0.4875 | 0.1063 | 0.4062 | 0.6906 |
>
> This result will also be updated into our paper . If you have further questions, please feel free to reply. Thank you again for your review and kind advice!

---

> ### Author Response · Authors · 2023-11-22
>
> Dear Reviewer,
>
> We would like to extend our sincere gratitude for your assistance and support. As time is limited, we wanted to confirm that our responses have addressed any concerns you may have had. If there are still any outstanding issues, please do not hesitate to let us know. We eagerly await your further feedback and hope that if all of your main concerns have been resolved, you would consider raising your score.
>
> Thank you once again for taking the time to review our paper.
>
> Best regards,
> The Authors

---

### Official Review · Reviewer_9wuG · 2023-10-31

**Soundness:** 4 excellent
**Presentation:** 3 good
**Contribution:** 3 good
**Rating:** 6
**Confidence:** 4

**Summary:**

This work proposes a data-driven automatic instruction selection method for fine-tuning language models. The basic assumption for data quality estimation is that it can be determined by the inference loss on the evaluation datasets. However it is expensive to have inference every time. Hence this work adopts several natural language indicators to predict the inference loss, i.e., the instruction data quality. By searching the best subset among the entire dataset, fine-tuned models over subsets can achieve state-of-the-art performance on two benchmarks.

**Strengths:**

Determining the quality of instruction data is complex and difficult. Also instruction data selection might be strongly dependent on downstream tasks. Data-driven methods like this work can be efficient to estimate the data quality from the proxy of inference loss on evaluation sets. As data selection can be a combinatorial optimization problem, this work provides a feasible data-driven solution and starts a good research question in this direction. It can be extended to other indicators as well.

**Weaknesses:**

1). The test set for rule fitting might not be well designed. This work samples instructions from the Self-Instruction dataset, which are most related to traditional NLP tasks. It is still confusing that learned Eq.4 can generalize well on more real-world scenarios like Figure (3)b.

**Questions:**

1). Why can the learned Eq.4 work well on unseen instruction datasets? Is there any intuitive explanations or any insight from the perspective of theory for example any theory guarantee of the proposed method?

2). This work only compared with random data selection methods as baselines. Are there any simple or straight-forward data selection methods for comparison?

**Details Of Ethics Concerns:**

n.a.

---

> ### Author Response · Authors · 2023-11-16
>
> Thank you for your review and for your thoughtful feedback! We are glad for your positive comments on the novelty and feasibility of our approach to select high-quality instruction-following data. Below is our response for your concerns and questions.
>
> - **Q.1 Why learned Eq.4 can work well on unseen instruction datasets? Is there any theory guarantee of the proposed method?**
>
> Thanks for pointing out the weakness in our paper. We don’t have a theory guarantee of our method currently, which is a little bit out of our scope. However, a possible reason of the effectiveness of our method is that, based on our current hypothesis, which are:
> 1. The rule-fitting test set is a high-quality dataset at the time of experiments.
> 2. Inference loss can indicate a model’s performance to some extent.
>
> Our method would possibly select data examples which are of similar quality to the selected rule-fitting set. Hence, for an unseen dataset, our proposed framework can:
> 1. Label data quality. Our proposed rule can help quantify and label example quality.
> 2. Search in an effective way and converge towards a certain direction. After we get the labels, we will rank the data according to their quality scores, and set a quantile point to choose the top high-quality examples. To do this, we use Flaml blendsearch to ensure that the final chosen dataset converges on a relatively low inference loss.
>
> We also would like to point out that the effectiveness of our method is based on the superficial alignment hypothesis[2] and mainly focuses on instruction tuning LLMs. We believe that base LLMs already have the knowledge to answer certain problems, while instruction tuning teaches the LLM to learn to understand and follow the instructions. Hence, we believe instruction tuning does not always require large scale training data. Through small amount of high-quality examples, an LLM should be able to learn to follow instructions.[1,2]
>
> - **Q.2 Other data selection methods?**
>
> Currently, instruction-following natural language data selection methods can be divided into three types:
> 1. Human selection. This method requires human efforts to label and select examples from datasets, which always results in large expenses.
> 2. Strong LLM labeling. This method requires serving or connecting very strong LLMs through apis, e.g., GPT-4. This method might lack explainability and also result in relatively large expense if the LLM api is not free and have high rate limits.
> 3. Machine selection. This method requires a model e.g., reward model, to mimic human or strong LLM to label data and set selection quantiles. This method can save efforts and work under very low expense.
>
> At the time of our submission, only (a) and (c) is available. For (b), since our resource is limited at this point, leveraging strong LLMs like GPT-4 to label over 100k data examples is too expensive for us. We will try our best to find other ways to implement (b).
> For (a), we are currently conducting experiments using LIMA data. We will provide the result as soon as possible and will update it into our paper and comment here. And for (c), please refer to our ablation study section.
>
> *Reference*:
>
> [1] Kung, Po-Nien, and Nanyun Peng. "Do Models Really Learn to Follow Instructions? An Empirical Study of Instruction Tuning." arXiv preprint arXiv:2305.11383 (2023).
>
> [2] Zhou, Chunting, et al. "Lima: Less is more for alignment." arXiv preprint arXiv:2305.11206 (2023).

---

> > ### Comment · Reviewer_9wuG · 2023-11-23
> >
> > Thanks for your response and they are helpful for clarification. Most of them answer my questions but I am still curious about the question I raised in the Weakness "This work samples instructions from the Self-Instruction dataset, which are most related to traditional NLP tasks. Why learned Eq.4 can generalize well on more real-world scenarios like Figure (3)b.?". It might need more investigations.

---

> ### Author Response · Authors · 2023-11-19
>
> Dear Reviewer 9wuG,
>
> We just got the result of LIMA-7b model (LLaMA-2-7B finetuned with LIMA data). Its OpenLLM benchmark scores are shown in the table below.
> |Model | Datasize | loss@self-instruct| loss@mt-bench| OpenLLM Avg| ARC| Hellaswag| MMLU|TruthfulQA|
> |---|---|---|---|---|---|---|---|---|
> |LIMA-7b|1000|0.9947|0.7604|0.5533|0.5563|0.8009|0.4371|0.419|
> |InstructMining-7b|1000| 0.9576 | 0.7114 | 0.562525  | 0.5589 | 0.7877 | 0.4299 | 0.4736 |
>
> We also compared its performance with our finetuned InstructMining model using LLM-as-a-judge. Results are listed in the table below.
> | Model | Win Rate | Lose Rate | Tie Rate | Adjusted Win Rate |
> |---|---|---|---|---|
> |InstructMining-7b(vs LIMA-7b)| 0.4875 | 0.1063 | 0.4062 | 0.6906 |
>
> This result will also be updated into our paper . If you have further questions, please feel free to reply. Thank you again for your review and kind advice!

---

### Official Review · Reviewer_rbEk · 2023-11-01

**Soundness:** 3 good
**Presentation:** 3 good
**Contribution:** 3 good
**Rating:** 5
**Confidence:** 4

**Summary:**

The paper proposed a linear combination of some indicators to predict model performance on an unseen instruction tuning dataset, and thus use this strategy to select useful new data to train the model. They also discovered a phenomenon in the transition of data quality and data quantity above a certain threshold.

**Strengths:**

1. Instruction selection is an important task as the number of annotations of instruction data continuously growing.
2. The paper provide a feasible way to predict model performance without training the model on the whole dataset, and thus using this method to select useful data.

**Weaknesses:**

1. I'm concerned about the expense of training the linear function. It requires training 129 LLMs on 129 subsets. My questions are how 129 is chosen and how long it takes to train the 129 LLMs?
2. The generalization ability of this method is unclear. There are two dimensions: the model used and the rule-fitting set for the combination function. I suppose the coefficients of indicators are model-dependent. The authors only tested on the Llama-2-7B model, I wonder whether using the same set of coefficients is possible for different models. Also, how sensitive the coefficients are towards the choice of the meta-training set, i.e., the rule-fitting set. Do you have numbers to demonstrate the stability of it?
3. I'm not sure whether using the inference loss as the y-axis of the double descent figure is the correct choice.

**Questions:**

1. Can you please answer Point 1 and 2 in the weakness section? Specifically, the total training time for the linear function. Your thoughts on the generalization ability across models and choice of meta-training set.
2. Just a reference. A recent work also considers task selection in instruction tuning. They use models' sensitivity to instructions to mine helpful tasks. This looks more efficient than the proposed method in this paper. They also have similar observation that when data size is enlarged, the effectiveness of selection method becomes compromised.
https://openreview.net/forum?id=DUJVphC9qR&referrer=%5Bthe%20profile%20of%20Po-Nien%20Kung%5D(%2Fprofile%3Fid%3D~Po-Nien_Kung1). How
3. Can you give a hypothesis on why there an increase in inference loss when the data selection is still more important than data quality? Does this mean the data selection is selecting harmful data also?
4. Why other baselines are even worse than random selection? Does this mean that you need more advanced baselines to compare with?

---

> ### Author Response · Authors · 2023-11-16
>
> Thank you for your review, and for recognizing our contributions! We are encouraged that you find our proposed framework feasible for selecting useful instruction following data. Below is our response to your questions and concerns:
>
> - **Q.1 & W.1: Expense of training.**
>
> In implementing regression with ten variables, it is imperative to maintain an Event Per Variable (EPV) ratio exceeding 10. In pursuit of analytical robustness, we intentionally surpassed this threshold by collecting over 100 experiments. Each subset comprising 1,000 samples was trained with 8 A100 GPUs within 15 minutes. As a result, the cumulative time for these 129 experiments totals approximately 32 hours.
>
> - **Q.1 & W.2: generalization of the rule**
>
> In table 6 we also show the generalization ability for our rule across model (LLaMA-1-7B), size(LLaMA-2-13B) and LoRA(LLaMA-2-7B with LoRA).
>
> - **Q.1 & W.2: coefficients sensitivity for rule-fitting set**
>
> Certainly, the coefficient has sensitivity to the test set for rule-fitting. However, it's crucial to note that such changes do not compromise the validity of our methodology. Selecting an ideal set of rule fits presents a challenge, and for our study, we employed a dataset generated through human-labeled GPT-4, representing the best available option at the time of publication.
>
> - **Q.2: Reference.**
>
> Thank you for pointing this reference out, As this reference link is not openable from our side. If you are referring to [Active Instruction Tuning](https://arxiv.org/abs/2311.00288) [1], we will cite this paper in our later version.
>
> - **Q.3: Why is there an increase in inference loss when data selection is more important than data quality?**
>
> Could you please provide more detail regarding “data selection is more important than data quality”?
>
> - **Q.4.1: Why are other baselines worse than random selection?**
>
> If you are referring to Table 4, there are many other results in the open LLM leaderboard. The OpenLLM leaderboard results only show one aspect of the model ability, hence its ability in question answering.
> We also would like to point out that, during data quality evaluation and quantile searching, we mainly use inference loss as the indicator of data quality, which means that our optimization goal is to find a training set which results in relatively low inference loss. This optimization goal is highly correlated with a model’s ability in conversation. However, OpenLLM leaderboard benchmarks are more focused on question answering. In this case, we cannot promise that optimizing using inference loss would result in better performance in question answering tasks.
>
> - **Q.4.2: Are there other baseline methods?**
>
> For other possible baselines, currently, instruction-following natural language data selection methods can be divided into three types:
> 1. Human selection. This method requires human efforts to label and select examples from datasets, which always results in large expenses.
> 2. Strong LLM labeling. This method requires serving or connecting very strong LLMs through apis, e.g., GPT-4. This method might lack explainability and also result in relatively large expense if the LLM api is not free and have high rate limits.
> 3. Machine selection. This method requires a model e.g., reward model, to mimic human or strong LLM to label data and set selection quantiles. This method can save efforts and work under very low expense.
>
> At the time of our submission, only (1) and (3) is available. For (2), since our resource is limited at this point, leveraging strong LLMs like GPT-4 to label over 100k data examples is too expensive for us. We will try our best to find other ways to implement (2).
> For (1), we are currently conducting experiments using LIMA data. We will provide the result as soon as possible and will update it into our paper and comment here. And for (3), please refer to our ablation study section.
>
> - **W.3: loss in double descent**
>
> Since loss is the main optimization objective in our paper, we use loss in y-axis in figure 1. We believe that loss is highly correlated with PPL, which used to be a commonly used metric for evaluating traditional generative language models. Hence, inference loss could be an indicator of a model’s performance. We’ve also done some experiments on other metrics. Please refer to Figure 4 for more details.
>
> *References*:
>
> [1] Kung, Po-Nien, et al. "Active Instruction Tuning: Improving Cross-Task Generalization by Training on Prompt Sensitive Tasks." arXiv preprint arXiv:2311.00288 (2023).

---

> > ### Author Response · Authors · 2023-11-19
> >
> > Dear Reviewer rbEk,
> >
> > We just got the result of LIMA-7b model (LLaMA-2-7B finetuned with LIMA data). Its OpenLLM benchmark scores are shown in the table below.
> > |Model | Datasize | loss@self-instruct| loss@mt-bench| OpenLLM Avg| ARC| Hellaswag| MMLU|TruthfulQA|
> > |---|---|---|---|---|---|---|---|---|
> > |LIMA-7b|1000|0.9947|0.7604|0.5533|0.5563|0.8009|0.4371|0.419|
> > |InstructMining-7b|1000| 0.9576 | 0.7114 | 0.562525  | 0.5589 | 0.7877 | 0.4299 | 0.4736 |
> >
> > We also compared its performance with our finetuned InstructMining model using LLM-as-a-judge. Results are listed in the table below.
> > | Model | Win Rate | Lose Rate | Tie Rate | Adjusted Win Rate |
> > |---|---|---|---|---|
> > |InstructMining-7b(vs LIMA-7b)| 0.4875 | 0.1063 | 0.4062 | 0.6906 |
> >
> > This result will also be updated into our paper . If you have further questions, please feel free to reply. Thank you again for your review and kind advice!

---

> > ### Comment · Reviewer_rbEk · 2023-11-22
> >
> > For Q.1 & W.2, I mean if your coefficients are obtained by training a, say, Llama-2-7B, is it still effective if you use this set of coefficients or Llama-1-7B? How about models from different families? For example, coefficients with Llama but instruction selection for T5. I'm a bit confused by what different base models mean in Table 6.
> >
> > For coefficients sensitivity for rule-fitting set, I'm still concerned how transferable this method is if the test set is OOD. You should have used a validation set for fitting the linear regression model. But what if the validation set is much different from the test set? Will that still effective?
> >
> > For Q.4.1, yes, it is Table 4. But I still don't quite get why Llama-2-chat or VICUNA will be worse than the INSTRUCTMINING-Random. Do you have the performance without any selection but simply use all the data from your two datasets? I just mean that the four models in the last four rows look too weak compared to your Instruction mining methods, although they use more data to do instruction tuning. I want to understand why you compare with these models and is that possible because the original datasets you select from have higher quality?
> >
> > For “data selection is more important than data quality”, apologies for the typo. I mean data size is more important than data quality. I saw some increase in the inference loss when the data size reached 80K for self-instruct and 60K-80K for mt-bench. Just curious why that might happen.
> >
> > I will keep my score for now.

---

> > > ### Author Response · Authors · 2023-11-23
> > >
> > > Thanks for your response. For your concerns, please find our responses below.
> > >
> > > - **Q.1 & W.2**
> > >
> > > Thanks for pointing this out. We are conducting new experiments on flant5-xl now and will soon share the results here.
> > >
> > > To be noticed, the reason we choose to use LLaMA-1-7b is that it shares a similar tokenizer with LLaMA-2-7b, where we did our fitting experiments. This makes the two results comparable. We choose to do the fitting experiments on LLaMA-2-7b since this is one of the most popular foundation model people are using, and we want our method to be useful for the community and others can adapt our method to save more efforts.
> > >
> > > For your second question, since the validation set, hence, rule-fitting set is chosen according to a certain task, which is instruction tuning in our paper, normally, the validation set will not be OOD with the test set. For example, if you want to use this method to select high-quality examples for machine translation, you will need to select a machine translation related validation set.
> > >
> > > If you are referring to an OOD test set which focuses on another kind of instruction based generation, e.g., coding tasks or writing tasks, we think our method is also able to generalize to those situations. From Figure 3(b), we can see that the model fine tuned with instruct-mining selected data can also improve a model’s ability in coding and writing. However, for a task which is much different from instruction tuning, we still recommend estimating a new Eq(3). Also, we would like to point out that our method focuses on instruction tuning, where the phenomenon, “less is more for alignment” is observed. For some other tasks, this observation might not stand.
> > >
> > > - **Q.4.1**
> > >
> > > Thanks for your comments, and we think this is really a good question that is worth further research. Here are some of our current observations and some speculation:
> > >
> > > *Q: Why models fine tuned with large scale datasets can sometimes perform worse than models finetuned on smaller datasets?*
> > >
> > > This difference might come from:
> > >
> > > - Double descent. As proposed in our paper, sometimes adding more data for finetuning does not always promise better results. This phenomenon is also observed in [5].
> > >
> > > - Evaluation set is not aligned with the optimization goal. The OpenLLM benchmark mainly focuses on question answering, hence, generating the correct answer is the focus of the benchmark. However, instruction tuning mainly focuses on generalizing LLMs to multiple tasks by enhancing its conversational ability. For finetuned models like Vicuna, it is possible that when enhancing the model’s conversational ability through finetuning on large-scale dataset, the model’s knowledge base begins to “shift”. During finetuning, the model’s ability in following instructions is improved, however, its ability to do question answering is harmed. This phenomenon also shows up when instruction tuning a code generation model. For example, in [1] table 4, when comparing the instruction tuned codellama with the original codellama, instruction tuning can sometimes hurt the model’s performance.
> > >
> > > *Q: Why do we choose these models as our baselines? Is it possible that our method generates better performance because we use data of higher quality?*
> > >
> > > We choose these models as our baselines since they are state-of-the-art models on the OpenLLM benchmark at the time of submission. Currently we can provide further results on model finetuned on full dolly dataset:
> > >
> > > |Model | Datasize | Loss@self-instruct | Loss@mt-bench | OpenLLM Avg| ARC| Hellaswag| MMLU|TruthfulQA|
> > > |---|---|---|---|---|---|---|---|---|
> > > |Dolly-full| 15,000| 1.047 | 0.807 | 0.546 | 0.566 | 0.808 | 0.437 | 0.371 |
> > >
> > > Additionally, we also finetuned a series of sharegpt based model. Below are the experimental results.
> > >
> > > | Model                 | ARC   | Hellaswag | MMLU  | TruthfulQA | Avg.  |
> > > | --------------------- | ----- | --------- | ----- | ---------- | ----- |
> > > | Sharegpt-selected-40k | 55.12 | 78.96     | 50.99 | 53.18      | 59.42 |
> > > | Sharegpt-selected-60k | 54.14 | 78.59     | 51.44 | 52.90      | 59.26 |
> > > | Sharegpt-full         | 55.72 | 80.94     | 47.47 | 48.34      | 58.11 |
> > > | Vicuna-v1.3           | 50.43 | 76.92     | 48.14 | 47.01      | 55.63 |
> > > | Vicuna-v1.5           | 53.24 | 77.39     | 51.04 | 50.34      | 58.00 |
> > >
> > > Please note that Vicuna uses a different version of ShareGPT from ours, while Vicuna uses more data, around 125k conversations from sharegpt.com.[2] We use the dataset from [3]. All models in the table are trained on sharegpt data.
> > >
> > > Also, you can refer to [4] for more OpenLLM benchmark results on other models.

---

> > > ### Author Response · Authors · 2023-11-23
> > >
> > > Dear Reviewer rbEk,
> > >
> > > Thank you for waiting. We've just got our results on Falcon-7B models, which is from a different foundation model family. The results are presented in the table below.
> > >
> > > | Model     | Data              | Loss@mt-bench | Loss@self-instruct |
> > > | --------- | ----------------- | ------------- | ------------------ |
> > > | Falcon-7B | Dolly-selected 1k | 1.1344        | 1.2780             |
> > > | Falcon-7B | Dolly-random 1k   |  1.2396  |  1.3723    |
> > > | Falcon-7B | Dolly-all         |   1. 3028  |   1.3960     |
> > >
> > > We finetuned Falcon-7b[1] using the selected 1000 dolly data, which we used in the paper to finetune LLaMA-2-7b. Also, we use the random 1000 data from the paper and full dolly data to finetune another Falcon-7B.
> > >
> > > We hope that our results can help resolve your concerns. If you have further questions, please don't hesitate to ask. Since we are approaching the deadline of the discussion phase, it would be really nice of you to raise your rating if most of your concerns have been resolved.
> > >
> > > *Reference:*
> > >
> > > [1] Falcon-7b model weight: https://huggingface.co/tiiuae/falcon-7b

---

> ### Author Response · Authors · 2023-11-22
>
> Dear Reviewer,
>
> We would like to extend our sincere gratitude for your assistance and support. As time is limited, we wanted to confirm that our responses have addressed any concerns you may have had. If there are still any outstanding issues, please do not hesitate to let us know. We eagerly await your further feedback and hope that if all of your main concerns have been resolved, you would consider raising your score.
>
> Thank you once again for taking the time to review our paper.
>
> Best regards,
> The Authors

---

> ### Author Response · Authors · 2023-11-23
>
> [Continued]
>
> - **Data size is more important than data quality**
>
> This shift could come from randomness and we have added some experiments for further reference. We apologize that we can only provide experiment results on datasets of relatively smaller size because of the limited time we have at this moment.
>
> | Datasize | Metric                    | Average | Std.           | Value       |
> | -------- | ------------------------- | ------- | -------------- | ----------- |
> | 1000     | avg/std of loss           | 0.981   | 0.01694909457  | 0.981±0.017 |
> | 1000     | avg/std of loss(mt bench) | 0.725   | 0.01788567698  | 0.725±0.018 |
> | 2000     | avg/std of loss           | 0.979   | 0.01174901483  | 0.979±0.012 |
> | 2000     | avg/std of loss(mt bench) | 0.713   | 0.001423670377 | 0.713±0.001 |
> | 3000     | avg/std of loss           | 1.002   | 0.0104352034   | 1.002±0.01  |
> | 3000     | avg/std of loss(mt bench) | 0.731   | 0.00483587947  | 0.731±0.005 |
> | 4000     | avg/std of loss           | 1.009   | 0.009194865843 | 1.009±0.009 |
> | 4000     | avg/std of loss(mt bench) | 0.744   | 0.0135088377   | 0.744±0.014 |
> | 5000     | avg/std of loss           | 0.996   | 0.004393212742 | 0.996±0.004 |
> | 5000     | avg/std of loss(mt bench) | 0.741   | 0.008913529242 | 0.741±0.009 |
> | 10000    | avg/std of loss           | 1.012   | 0.008364151272 | 1.012±0.008 |
> | 10000    | avg/std of loss(mt bench) | 0.750   | 0.007255154838 | 0.75±0.007  |
> | 40000    | avg/std of loss           | 1.015   | 0.003903844259 | 1.015±0.004 |
> | 40000    | avg/std of loss(mt bench) | 0.740   | 0.03513605203  | 0.74±0.035  |
>
> In this table, we can see that a significant change in loss should be larger than ~0.005 to ~0.01, while the difference between Select-loss and Random-loss are too small. Generally, if we focus on the general trend of the two lines in Figure 1, we can see that after around 40,000 to 50,000, the difference between Select-loss and Random-loss becomes smaller, which means that the importance of data selection decreases.
>
> Thanks for your response again. Your review is of great help for improving our paper. If you feel most of your concerns have been addressed, it would be very nice of you to raise the score.
>
> *Reference:*
>
> [1] Roziere, Baptiste, et al. "Code llama: Open foundation models for code." arXiv preprint arXiv:2308.12950 (2023).
>
> [2] Vicuna-v1.5 model card: https://huggingface.co/lmsys/vicuna-7b-v1.5
>
> [3] Sharegpt unfiltered dataset: https://huggingface.co/datasets/anon8231489123/ShareGPT_Vicuna_unfiltered
>
> [4] OpenLLM Leaderboard: https://huggingface.co/spaces/HuggingFaceH4/open_llm_leaderboard
>
> [5] Nakkiran, Preetum, et al. "Deep double descent: Where bigger models and more data hurt." Journal of Statistical Mechanics: Theory and Experiment 2021.12 (2021): 124003.

---

### Meta-Review · Area_Chair_Wkym · 2023-12-17

**Metareview:**

While the paper presents an interesting study on data selection for fine-tuning language models via instruction mining, some concerns cast doubt on the impact and novelty of the presented approach. The methodology primarily hinges on measuring dataset quality by inference loss on an evaluation dataset, which is a contested method and may not universally indicate better actual performance on downstream tasks. Reviewers questioned whether such measures translate reliably to practical enhancements in various NLP tasks, particularly given the lack of gold standard outputs in instruction tuning contexts. Despite the authors' responses and additional experiments, the uncertainties around the practical implications of the described loss measurements persist. Multiple reviewers also raised the issue of potential overfitting to the specific evaluation dataset, emphasizing the limitations of this evaluation strategy.

After careful consideration of the paper, reviewers’ feedback, and the author's rebuttal and additional comments, the reviewers reached an agreement that the paper is not ready for the conference.

**Justification For Why Not Higher Score:**

The methodology primarily hinges on measuring dataset quality by inference loss on an evaluation dataset, which is a contested method and may not universally indicate better actual performance on downstream tasks. Reviewers questioned whether such measures translate reliably to practical enhancements in various NLP tasks, particularly given the lack of gold standard outputs in instruction tuning contexts. Despite the authors' responses and additional experiments, the uncertainties around the practical implications of the described loss measurements persist. Multiple reviewers also raised the issue of potential overfitting to the specific evaluation dataset, emphasizing the limitations of this evaluation strategy.

**Justification For Why Not Lower Score:**

N/A

---

### Decision · Program_Chairs · 2024-01-16

Reject